# Integration of landmark and saccade target signals in macaque frontal cortex visual responses

Adrian Schütz[1,2,5], Vishal Bharmauria [3,5], Xiaogang Yan[3], Hongying Wang[3], Frank Bremmer [1,2,6] &
J. Douglas Crawford [3,4,6 ✉]

Visual landmarks influence spatial cognition and behavior, but their influence on visual codes for action is poorly understood. Here, we test landmark influence on the visual response to saccade targets recorded from 312 frontal and 256 supplementary eye field neurons in rhesus macaques. Visual response fields are characterized by recording neural responses to various target-landmark combinations, and then we test against several candidate spatial models. Overall, frontal/supplementary eye fields response fields preferentially code either saccade targets (40%/40%) or landmarks (30%/4.5%) in gaze fixation-centered coordinates, but most cells show multiplexed target-landmark coding within intermediate reference frames (between fixation-centered and landmark-centered). Further, these coding schemes interact: neurons with near-equal target and landmark coding show the biggest shift from fixation-centered toward landmark-centered target coding. These data show that landmark information is preserved and influences target coding in prefrontal visual responses, likely to stabilize movement goals in the presence of noisy egocentric signals.

[1] Department of Neurophysics, Phillips Universität Marburg, Marburg, Germany. [2] Center for Mind, Brain, and Behavior – CMBB, Philipps-Universität Marburg, Marburg, Germany & Justus-Liebig-Universität Giessen, Giessen, Germany. [3] York Centre for Vision Research and Vision: Science to Applications Program, York University, Toronto, Canada. [4] Departments of Psychology, Biology, Kinesiology & Health Sciences, York University, Toronto, Canada. [5] These authors contributed equally: Adrian Schütz, Vishal Bharmauria. [6] These authors jointly supervised this work: Frank Bremmer, J. Douglas Crawford. ✉email: jdc@yorku.ca

In daily life, we often use visual landmarks for navigation and goal-directed behavior[1–3]. In the case of goal-directed movements, it is thought that stationary landmarks provide allocentric cues to stabilize the noisy egocentric transformations from sensory inputs (e.g., relative to the eye) to motor commands relative to the head or body[4–8]. For example, the edge of a desk might provide additional cues for grasping a coffee mug on its surface. Various theories have been postulated for the integration of landmarks into egocentric codes for action[2,9–14]. However, the neural mechanisms are poorly understood. Specifically, it is not known if landmark information is integrated with action goals in the visual system and then relayed to the motor system, or if this occurs within the motor system.

It has been speculated that the visual system initially codes this influence as target-landmark configurations (i.e., the spatial locations of the target-relative-to-the-landmark)[2,15]. Human neuroimaging studies suggested that egocentric and allocentric visual codes are separated in the dorsal and ventral visual streams, respectively[16–18] and then converge in the frontal cortex for action[19]. At the cellular level, visual response fields (the area of space where stimuli influence neural activity) are modulated by the presence of other stimuli (e.g., landmarks or distractors) within or outside of a response field[20,21]. In particular, it has been reported that visual landmarks and other surrounding distractors influence neural activity in the superior colliculus[22], parietal cortex[23,24], and precuneus[25], sometimes causing spatial shifts in response fields[22–25]. However, these studies did not specifically investigate the integration of ego/allocentric coordinates.

When egocentric and allocentric cues conflict, the brain appears to optimally weigh these cues to derive the best estimate for an accurate movement[9,10,26–29]. For example, when monkeys made head-unrestrained gaze shifts toward remembered targets in the presence of a landmark shift, gaze end points also shifted partially (~1/3) in the same direction[7,30,31]. This ego/allocentric weighting might reduce gaze errors by improving the internal estimates of initial three-dimensional (3D) eye orientation, which is more variable in natural head-unrestrained conditions[32,33]. Failure to compensate for torsional tilts of the retina will lead to errors in aiming gaze and reaching movements[34–36].

In a recent series of studies, we combined a memory-delay cue-conflict saccade task in head-unrestrained monkeys with 3D behavioral measures and neural recordings from the frontal (FEF) and supplementary (SEF) eye fields. In the absence of a landmark, FEF neurons showed a progressive transition from eye-centered saccade target coding in the visual response to eye-centered gaze coding (i.e., future gaze relative to initial eye orientation) in the saccade motor response[37,38]. When we introduced a large visual landmark (Fig. 1a) and shifted it during the memory delay (Supplementary Fig. 1), this immediately caused the neural code for target memory to shift in the direction of the landmark shift. This shift later became integrated in the peri-saccadic motor response for gaze behavior[30,31]. We proposed that this provides a neural signature for the ego/allocentric integration observed at the behavioral level.

However, it remains unclear how and where visual signals from sationary landmarks influence the gaze system. In our previous study, the FEF/SEF visual response was dominated by eye-centered target codes, even in the presence of a landmark[30,31]. However, in our cell population analysis, we pooled data across all cells and landmark configurations, and only tested 'cardinal' egocentric and allocentric models. It thus remains possible that visual landmarks produce more subtle effects on prefrontal visual responses, such as cell-specific effects, configuration-specific effects, and intermediate codes (i.e., between coding targets versus landmarks and between different reference frames).

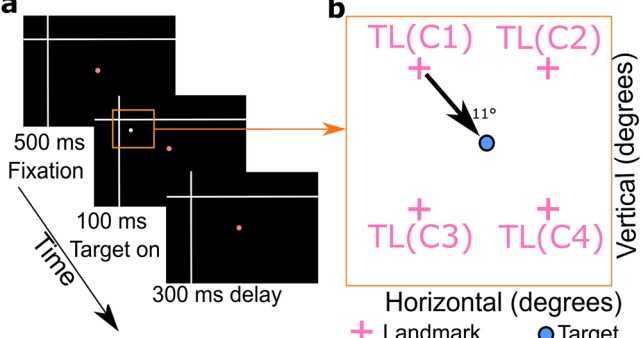

**Fig. 1 Experimental paradigm with different target-landmark configurations. a** Temporal sequence of stimulus presentation (fixation dot, landmark, target). The head-unrestrained monkey starts the trial by fixating a central red dot for 500 ms in the presence of two white intersecting lines (landmark). Then a white dot (target) is flashed (100 ms) in one of four possible locations relative to the landmark, followed by a 300 ms delay. The remaining parts of the paradigm (not analyzed here) are presented in the Supplementary Fig. 1. **b** Schematic of the four possible target-landmark configurations (*TLC1-4*). The orange square roughly corresponds to the orange region shown in (**a**) and does not include the fixation point.

Here, we examined the influence of a stable background landmark on the prefrontal visual response to saccade targets, using an expanded dataset from the same series of experiments (Fig. 1a), and a much more detailed analysis. Specifically, we asked (1) is landmark information preserved at the single cell level in prefrontal visual activity? (2) do stable visual landmarks influence intermediate reference frames for saccade target codes? (3) does this influence depend on specific target-landmark configurations? And (4) do landmark and target signals interact within prefrontal cells? The results show that landmark signals persist in prefrontal target responses, and these responses interact to generate target codes in intermediate ego/allocentric reference frames. This suggests that frontal cortex is an important site for integrating ego/allocentric visual cues for action.

## Results

**Experimental approach and general observations.** In this study, we investigated the influence of a static landmark on visual responses in two cortical gaze control areas, the FEF and SEF. Figure 1 shows the visual stimuli that were present before and during the neural responses analyzed in the current study. The entire paradigm is shown in Supplementary Fig. 1, including later response periods that we described in previous studies[7,30,31]. Figure 1a shows an example stimulus configuration where a background landmark (*L*: a large 'cross') first appears, followed by a transient (100 ms) appearance of the target (*T*). This landmark could appear in one of four spatial (oblique positions) configurations (*TLC1-4*) relative to the target stimulus (Fig. 1b). Later, after a delay, monkeys were rewarded for looking at the target stimulus (Supplementary Fig. 1). Note that there was no explicit reward for attending to the landmark, i.e., the target reward window (radius 8–12°) was large enough to neither reward/ punish any implicit landmark influence on behavior.

We found that there was a systematic bias (27% for Monkey L and 15% for Monkey V) of the final gaze location (gaze end point where the monkey's eye landed after the saccade was performed to the memorized target) toward the landmark, but the final gaze position correlated better with the target location (0.72 for Monkey L and 0.68 for Monkey V) compared with the landmark location (0.11 for Monkey L and 0.06 for Monkey V) as

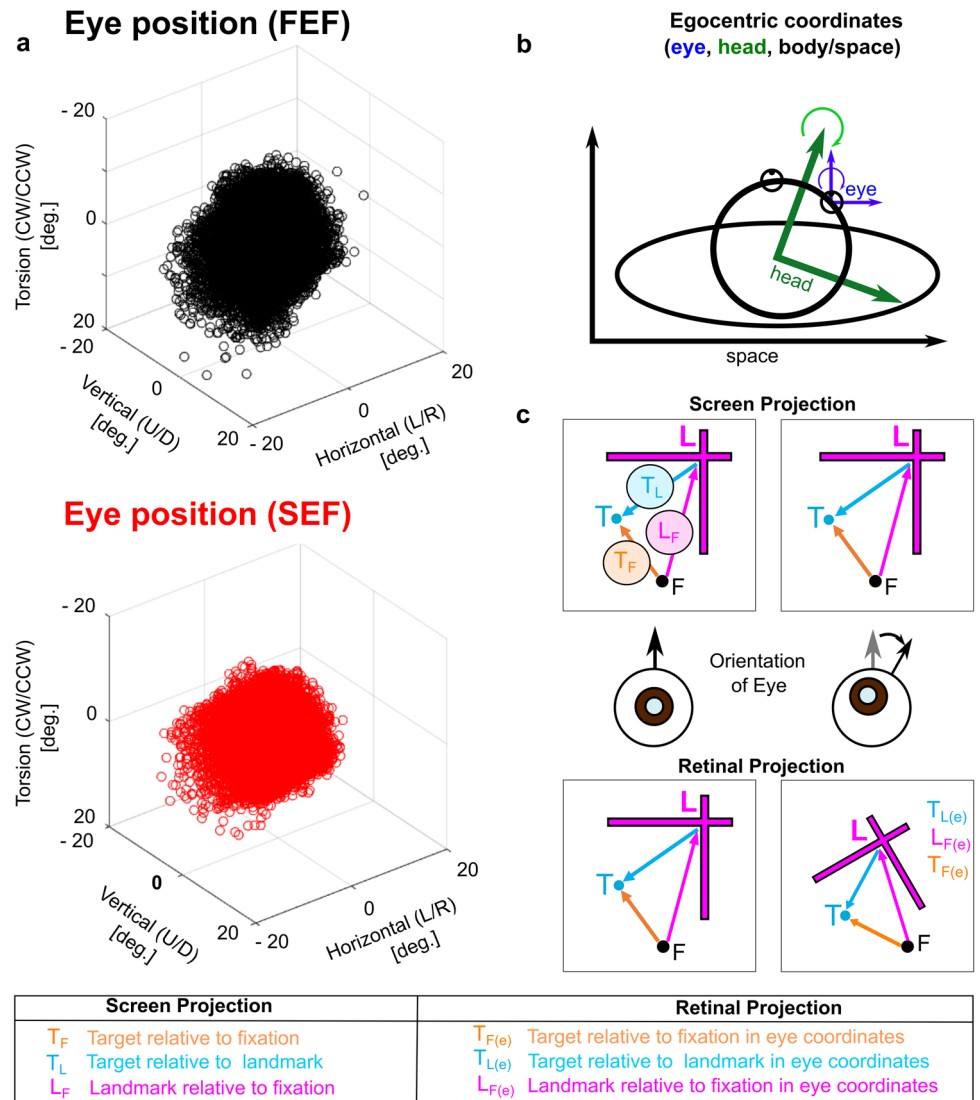

**Fig. 2 Geometric relationship between 3D eye orientation and spatial models. a** Scatter plot of the initial 3D eye orientation (in space) at the fixation (*F*) position. Data from every trial used in the analysis of FEF (top panel) and SEF (lower panel) neurons are shown for both monkeys. Note that in natural head-unrestrained conditions, eye-in-space torsion (vertical axis) is much more variable than in head-restrained conditions, causing the retina to rotate relative to space. **b** Schematic of the different egocentric coordinate systems available in this task. Curving arrows indicate rotation about the torsional axes of the eye and head. Note that in head-unrestrained conditions, eye orientation in space is determined by both the eye position relative to head, and the head relative to body. **c** Schematic of how eye torsion in space influences projection of the screen stimuli onto the retina for an example Target-Landmark Configuration (*TLC2*) relative to fixation (*F*). The top row represents the vectors between these stimuli ($T_F$, $T_L$, $L_F$) as they appear on the screen, the middle row shows a schematic of two initial eye orientations (primary position vs. torsionally tilted), and the bottom row shows the resulting retinal projections of those vectors in eye coordinates [$T_{F(e)}$, $T_{L(e)}$, $L_{F(e)}$], but without optic reversal, e.g., vectors on the left correspond to leftward stimuli. At primary (straight ahead) eye position, the retinal projections are relatively simple, but torsion results in both tilts and distortions of various stimulus vectors, depending on the different *T-L-F* configurations. These configurations could help remember where targets are, and the systematic distortions could help decode their locations in space in the presence of a faulty or noisy internal estimate of eye torsion in space. (Note that the amount of distortion has been slightly exaggerated in the figure for demonstrative purposes).

previously reported[7]. Finally, to capture the natural complexity of gaze behavior, animals viewed these stimuli head-unrestrained.

As noted above, eye-in-space orientation is more variable in natural head-unrestrained conditions, including considerable torsion of the eyes around the line of sight[32,33]. The distributions of initial 3D eye orientation in our two monkeys (recorded simultaneously with FEF and SEF recordings) are shown in Fig. 2a, resulting from variations in both initial eye and head orientation (shown schematically in Fig. 2b). This torsion in turn causes tilts and distortions of non-foveal stimuli such as those used here (Fig. 2c). Estimating and compensating for these distortions based on egocentric cues is a challenge for the visual

system[34–36,39], but the landmark provides additional visual cues (i.e., based on the prior assumption that vertical and horizontal orientations of the landmark are stable on the computer screen; the tilted image must be due to eye rotation). Conversely, these trial-to-trial variations are experimentally useful for dissociating ego/allocentric reference frames (see analysis methods below).

To account for these factors, 3D eye orientation was recorded and used in the calculation of all eye-centered target directions[33]. This allowed us to precisely calculate and contrast the retinal projections of the Target in eye coordinates [$T_{F(e)}$], where '*T*' designates the coded *parameter* (here the target), '*F*' designates the 2D '0,0' *coordinate system origin* (here gaze fixation/fovea),

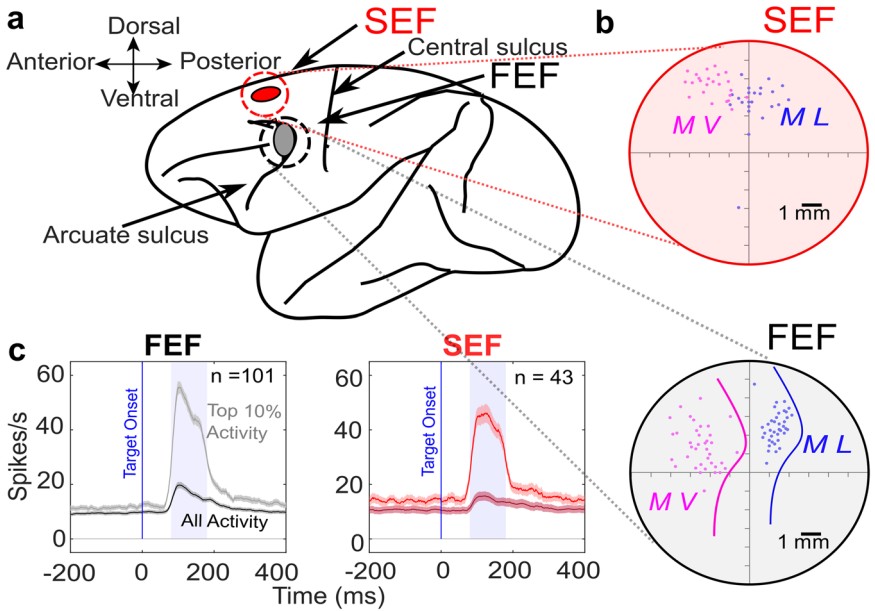

**Fig. 3 Simultaneous electrophysiological recordings from frontal and supplementary eye fields. a** The gray ellipse represents the location of the FEF, the red ellipse represents the location of the SEF. **b** Blow-up of the recording chambers in (**a**). The connected red and black disks represent the coordinates of the recording chambers, showing the sites (colored dots) of neural recordings for both monkeys [Monkey L (ML), Monkey V (MV)], also confirmed by microstimulation-evoked eye movements. Colored lines indicate the location of the arcuate sulcus within the FEF recording chamber for both monkeys. **c** Mean (± SD) of the spike-density plots of the visual responses for all FEF (gray) and SEF neurons (red) analyzed in this study. The darker trace represents the mean response of all trials (including low and high responses for all targets), whereas the lighter trace represents the top 10% responses for each neuron, usually corresponding to the 'hot' spot of the neuron's response field. The shaded areas show the temporal windows used for sampling data to quantify the visual response (ranging from 80–180 ms after the target onset).

and 'e' designates the 3D *reference frame* (here 3D eye orientation). This nomenclature is used to clearly distinguish this model from other potential coding schemes such as Landmark-relative-to-gaze fixation in eye coordinates [$L_{F(e)}$], and Target-relative-to-landmark in eye coordinates [$T_{L(e)}$]. In other potential coding schemes where the 0,0 is an arbitrary 'straight ahead' in lab coordinates, we only label the coded parameter and reference frame, e.g., Target in head [$T_{(h)}$] and Target in space [$T_{(s)}$][30,31,37].

During neural recordings, targets were randomly presented throughout each neuron's response field, while randomly varying the relative landmark configuration, providing a complete dataset for 312 FEF and 256 SEF neurons (Fig. 3a) from two monkeys (L and V). Of these, 154 FEF and 59 SEF neurons showed significant visual responses to the target relative to pre-stimulus baseline (as well as other later responses, not shown here). Figure 3b shows the blow-up of the recording sites inside the recording chamber for both monkeys. These neurons sometimes produced weak, sluggish responses to landmark onset (Supplementary Fig. 2), but these responses were untuned and partially dissipated before saccade target onset. Here, we only analyzed the response fields corresponding to the robust initial visual response to the target, quantified as the number of action potentials within a fixed temporal window after target presentation (Fig. 3c).

**Model fitting approach.** As in our previous studies, we used a model-fitting method[31,37,40,41], illustrated schematically in the left column of Fig. 4. This method allows one to compare neural activity against various 'cardinal' spatial models derived from the experimental measures of stimulus location, eye orientation, and head orientation (see "Methods"), such as Target-relative-to-Fixation [$T_{F(e)}$] and Target-relative-to-Landmark [$T_{L(e)}$] (Fig. 4a), again using measures that compensate for the 3D eye orientation

in space (Fig. 2). This method uses the same principle used in the other reference frame studies[42–47] but generalized to work for any spatial model in the presence of variable spatial parameters[40]. Here, these variations arise from variable 3D eye orientations (Fig. 2a) and different target-landmark configurations in our pooled data (Figs. 1b, 2c). In brief, non-parametric fits were made to the visual response as a function of two-dimensional target location, defined in the coordinates of each specific spatial model, e.g., $T_{F(e)}$, $T_{L(e)}$, etc. (Fig. 4b). The use of a non-parametric fit makes this procedure relatively immune to response field idiosyncrasies[40]. For each data point (trial), the neural activity is compared with a fit made to all the other data points. For example, if data points fit better in $T_{F(e)}$ coordinates, then $T_{F(e)}$ is the best model for that neuron and vice versa, i.e., the model that yields the lowest residuals (between the fit and actual data) is deemed the 'best' (Fig. 4b).

An example response field fit to an FEF visual neuron is shown in Fig. 4c–e, in this case the $T_{F(e)}$ model yielded the lowest residuals. Figure 4c provides the raster and spike density plot for all (black lines) and the top 10% (gray lines) of the pooled responses. This top 10% roughly corresponds to the 'hot spot' of the response field as typically defined. The gray shaded area (80–180 ms aligned to the target onset) indicates the temporal sampling window used for the response field fits. Figure 4d shows the corresponding non-parametric fit of the visual response field in the best coordinate frame [$T_{F(e)}$], where the origin corresponds to the fovea. The red area represents the hot-spot of the response field.

Figure 4e shows the actual data (black circles, sized in proportion to the response size for each trial), superimposed on the non-parametric fit, and with the residuals between the data and the fit are plotted on the right side. Some variability (residuals) generally persists even at the best fit coordinate frame, likely due to non-spatial factors such as attention and

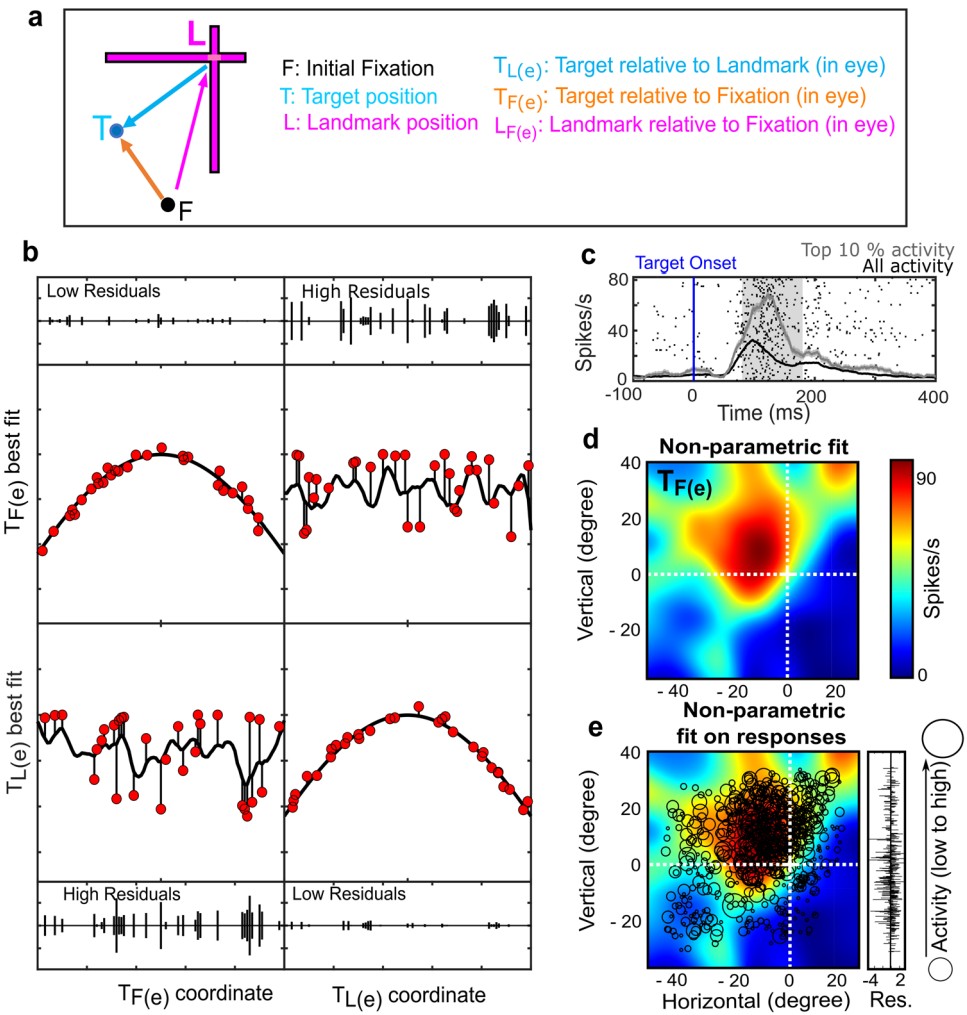

**Fig. 4 Method for fitting spatial models to neural response fields. a** Summary of the main models studied in this paper [$T_{F(e)}$, $T_{L(e)}$, $L_{F(e)}$] for an example *T-L-F* configuration (see key for details). **b** Schematic of the logic behind the response field analysis, using simulated data. The x-axes represent the spatial coordinate. The y-axes show neural activity. Neural responses from individual trials are represented by the red dots. The black curving lines show the non-parametric fits, which in practice did not restrain the response field to a specific (e.g., gaussian) shape. Above and below these plots are the residuals of fit, i.e., the difference between individual responses and the fit. In the left column, data and fits are plotted in the $T_{F(e)}$ coordinate system, whereas the right column plots data and fits in the $T_{L(e)}$ coordinate system. The upper row shows hypothetical activity from a neuronal response field when the target is coded relative to fixation, $T_{F(e)}$. In this case, low residuals result when the data are plotted and fit in the same coordinate, i.e., $T_{F(e)}$ (left), and high residuals result when the data are plotted relative to the landmark, i.e., $T_{L(e)}$ (right). The opposite pattern occurs in the panels in the bottom row for a neuron that encodes targets relative to the landmark [$T_{L(e)}$]. **c–e** Example analysis of a visual neuron with lowest residuals for the $T_{F(e)}$ model. **c** Raster and spike density plot of the neuron's activity. The blue vertical line indicates the target onset, and the gray shaded area represents the time window (80–180 ms after the target onset) used to quantify the visual response to the target. The raster shows the action potentials for trials with the top 10% activity (corresponding to the 'hot spot' of the response field) in the analysis window. The gray line shows the spike density for these same top 10% trials whereas the black line shows the spike density for all the trials. The confidence intervals show the standard error. **d** The non-parametric fit of the neuron's response field in the coordinates of its best model fit, $T_{F(e)}$ (with 0,0 origin at the intersection of the white cross, i.e., the fovea). Shown to the right is the color scale for the fit, ranging from low (blue) to high (red) activation. **e** Actual data (black circles placed at the stimulus location and scaled to the response in the sampling window—see key on right) plotted over the fit in the best fit coordinate system, with the difference between them (residuals) shown on the right.

motivation[8,30,31,48]. Unless otherwise stated, these response field data were only included in analysis if they showed significant spatial tuning, i.e., if the best fits had significantly lower residuals than 100 reconstructed response fields with shuffled neural activity relative to stimulus locations (see Supplementary Fig. 3 for an example and "Methods" for details).

**FEF and SEF populations statistics: predominance of $T_{F(e)}$.** In our previous studies[30,31], the target-in-eye model [here called $T_{F(e)}$] provided the best overall fit to our population of visual response fields. We did not probe this any further because those

studies focused on later memory and motor responses to the landmark shift. To reexamine the visual code here, we derived a dataset from the same experiments[30,31], but this new dataset was larger, because we did not need to remove trials with motor errors later in the task. The result was 101 FEF and 43 SEF neurons with significant response field tuning.

We began the current analysis by confirming the findings from our previous studies[30,31]. Specifically, we compared six potential visual models including the *target* relative to eye [$T_{F(e)}$], head [$T_{(h)}$], space [$T_{(s)}$], or landmark [$T_{L(e)}$], and landmark relative to the eye [$L_{F(e)}$] or space [$L_{(s)}$]. The fits were made to the visual response fields of each neuron, pooling across all landmark

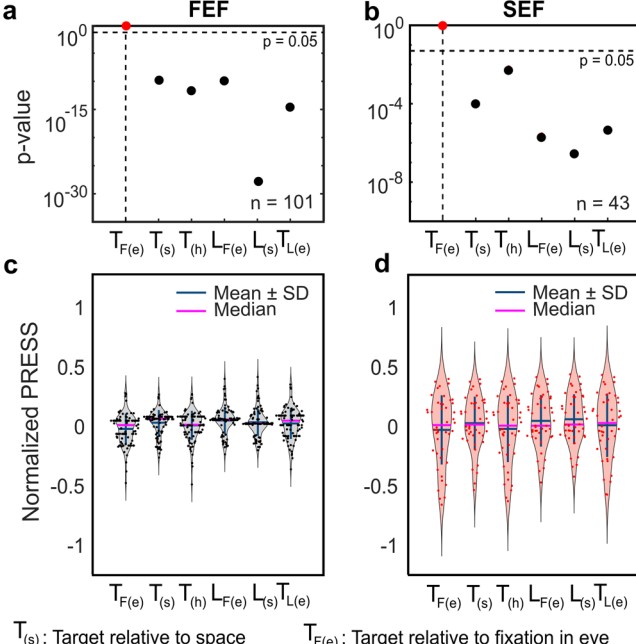

$T_{(s)}$: Target relative to space
$T_{(h)}$: Target relative to head
$L_{(s)}$: Landmark relative to space

$T_{F(e)}$: Target relative to fixation in eye
$L_{F(e)}$: Landmark relative to fixation in eye
$T_{L(e)}$: Target relative to landmark in eye

**Fig. 5 Statistical comparison of models at population level: $T_{F(e)}$ is the favored model in both FEF and SEF.** Top row: the model that yields the lowest residuals (red) is used as a statistical reference, and thus gives $p$ value of 1.0 relative to itself. Data points below $p = 0.05$ (the horizontal dashed line) indicate a model that gave significantly higher residuals, i.e., poor fits. **a** The $p$ value statistics and comparison between different models for all FEF neurons ($n = 101$). In this case, $T_{F(e)}$ gave the best fit and all other models were significantly worse. **b** Same as (**a**) but for SEF neurons ($n = 43$). $T_{F(e)}$ is the best model overall for both sites, with all other models statistically eliminated. **c** Normalized mean PRESS ($\pm$SD) residuals for FEF and **d** SEF neurons. The values for models were normalized by dividing by the mean PRESS residuals of the best fit, i.e., $T_{F(e)}$. Model Definitions: $T_{F(e)}$: Target-relative-to-fixation in eye coordinates (note that this is the same model as '$T_e$' in our previous publications; $L_{F(e)}$: Landmark-relative-to-fixation in eye coordinates; $T_{L(e)}$: Target-relative-to-landmark in eye coordinates; $T_{(s)}$: Target relative to space; $T_{(h)}$: Target relative to head; $L_{(s)}$: Landmark relative to space.

configurations, to calculate the mean PRESS residual for each fit[40]. These means were then used for statistical comparisons across the entire population of FEF and SEF neurons, as previously described[30,31]. This gave the same result that we reported previously: as summarized in Fig. 5a–d, both structures showed significantly lower residuals for the $T_{F(e)}$ fits compared with any of the other fits tested. Based on this analysis alone, one might again conclude that these visual responses only code the saccade target (relative to the eye) and show little or no influence of the landmark[30,31]. But here, we pursued a more detailed analysis of landmark influence, as described in the following sections.

**Fits to individual cells: heterogenous target and landmark coding.** In the current study, we examined and compared the best fits of the individual cells. Figure 6a, b shows a representative FEF response field which gave the lowest residuals when plotted in $T_{F(e)}$ coordinates (Fig. 6a). This fit was significantly better than $L_{F(e)}$ and other models with landmark terms (Fig. 6b). In this case, within-cell statistics were performed using the PRESS residuals for individual trials[40].

However, not all cells followed this trend. An example of a landmark favoring neuron, $L_{F(e)}$, is shown in Fig. 6c. In fact, for this neuron, all other models gave significantly higher residuals (poorer fits) (Fig. 6d). Overall, nearly 40% of FEF neurons preferred $T_{F(e)}$, but best fits for other neurons were also distributed across the other models (Fig. 6e), with $L_{F(e)}$ being a close second (~30% of FEF neurons preferring this). This suggests that, despite the weak initial landmark response (Supplementary Fig. 2) some FEF responses code landmark location.

We performed the same analysis on SEF response fields. An example of this analysis for an SEF neuron is presented in the Supplementary Fig. 4a–c. SEF neurons were generally more broadly tuned, however, due to the fragmented, scattered nature of 'hot spots' in some FEF and many SEF neurons, it was difficult to derive a simple measure (such as bandwidth) to compare their response fields. Overall, in SEF, ~40% of neurons preferred $T_{F(e)}$, however, other models were also well represented (Fig. 6f), such as Landmark relative to the eye (4.5%) and Target relative to the landmark (14%) or head (18%).

Overall, these results suggest that while many individual FEF and SEF cells agree with the population statistics (where target coding in eye coordinates dominates), some code other parameters, including landmark location (especially in FEF).

**Spatial continuum analysis.** So far, our analysis has only contrasted cardinal models, like $T_{F(e)}$, $L_{F(e)}$, and $T_{L(e)}$ (Figs. 5, 6). It is also possible that neural response fields utilize intermediate codes between these cardinal models, which would then be artificially forced into different cardinal categories in our previous analysis. To test this possibility, we developed two new spatial continua and a new method for determining the best fits along these continua. Figure 7, top row (a, b) illustrates the basic concept of the method, and the details are provided in the "Methods" section.

Figure 7a illustrates the two spatial continua that we used to test our hypotheses (preliminary analysis showed these to yield the clearest results). The first (the *T-L Parameter Continuum*) provides intermediate coding schemes between our main parameter codes (Target and Landmark) in the most prominent (fixation-centered) coordinate frame. To construct this, we created a mathematical continuum with 10 intermediate steps between $T_{F(e)}$ and $L_{F(e)}$. The second (the *F-L Coordinate Continuum*) utilizes similar steps between target coding in fixation-centered [$T_{F(e)}$] and landmark-centered [$T_{L(e)}$] coordinates. Since this required mathematical comparisons between arbitrary frames, we developed a new method based on response field weight modulation (see "Methods"). We then fit our data against each point along these continua, searching for the point that yielded the lowest residuals. For example, Fig. 7b shows example mean residuals obtained at each step of the *F-L coordinate continuum* for the response field shown in Fig. 4c–e, in this case showing the lowest residuals (best fit) exactly at $T_{F(e)}$.

Another methodological concern is that these continuum fits might be noise sensitive, i.e., the fit difference between each step might be small compared with noise in the data. To account for this possibility, we created a control dataset by shuffling target-landmark configuration 100 times relative to the neural activity/target location pair (see "Methods" for details). In this way, landmark spatial information (position relative to both target and fixation) was randomized in each response field without changing the target response field. The expectation is that noise would persist after shuffling, whereas the meaningful signal related to landmark location or target-landmark configuration should be lost.

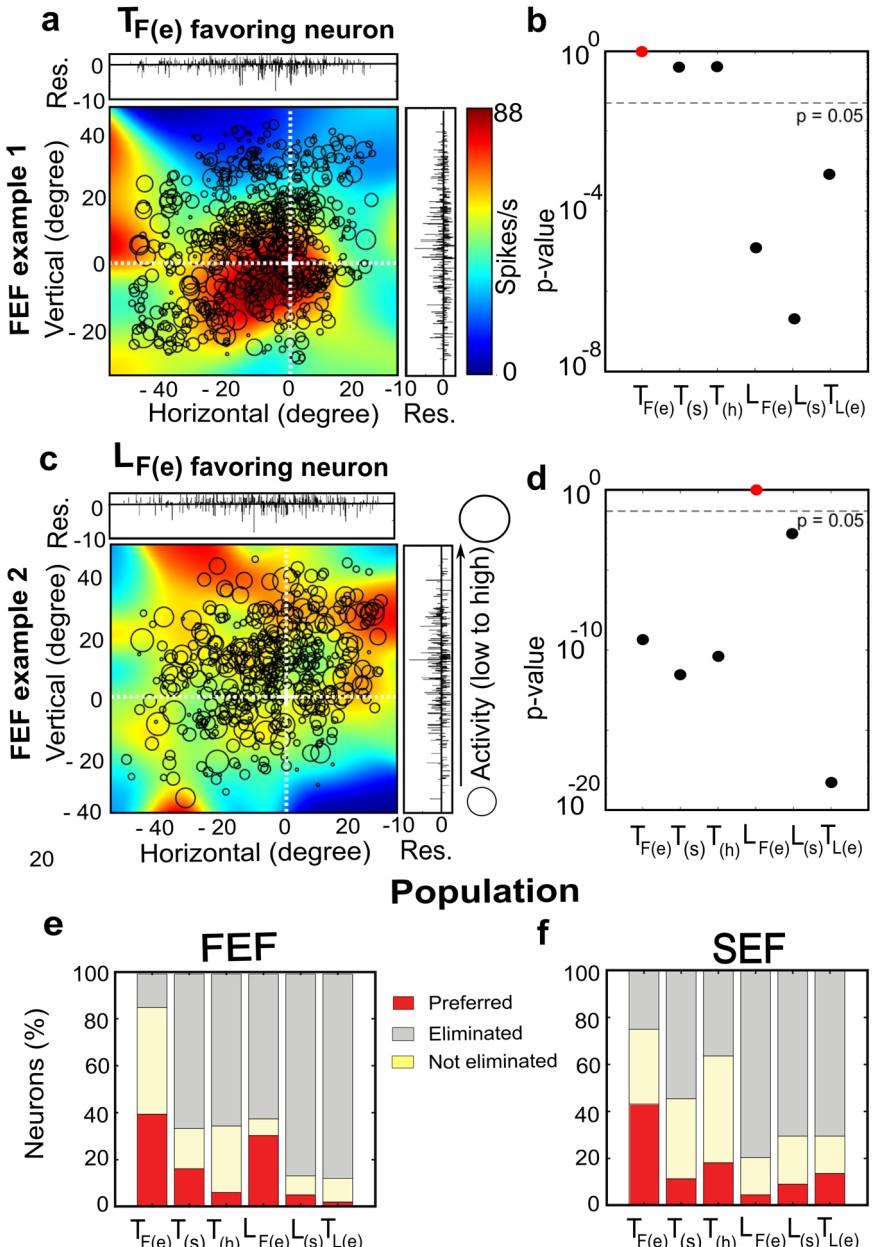

**Fig. 6 Typical neuron examples (FEF) and population analysis for FEF and SEF.** First two rows **a–d**: analysis of two example FEF neurons. **a** $T_{F(e)}$-preferring neuron plotted in eye coordinates. This neuron has a clear 'hot spot' to the left of the fovea (0,0). **b** The statistical analysis shows that compared with this model (red dot) most other models were statistically eliminated. **c** 'Opposite' example of a neuron that showed significantly lower residuals in $L_{F(e)}$ coordinates. In this case, the response field was more scattered with multiple hot-spots, and **d** all other models were statistically eliminated. Bottom row **e**, **f**: Percentage distribution of neurons in **e** FEF and **f** SEF. Best fits to each of the models are described in text and figure legend 5. Target-relative-to-eye [$T_{F(e)}$] was the most 'popular' model in both structures, but in FEF landmark-relative-to-eye [$L_{F(e)}$] was a close second. In other words, at least FEF was still coding landmark information.

The remainder of Fig. 7c–h shows our continuum analyses for FEF with the *T-L Parameter Continuum* in the middle row and the *F-L Coordinate Continuum* in the lower row (similar but more modest SEF results are provided in Supplementary Fig. 5). In each panel, best fit scores for each response field are plotted as a function of the corresponding shuffled control data. We hypothesized that if the landmark influence is real, the original dataset should be significantly different from the shuffled dataset. This could either take the form of individual neurons varying significantly from the main diagonal (increased spread) or an overall population shift above or below the diagonal. Results are discussed below.

**Target-landmark multiplexing: the *T-L* parameter continuum.** To test if the target and landmark information is multiplexed in visual responses at the level of individual neurons and populations, we performed model fits along the *T-L Parameter Continuum*. The results for FEF response fields are shown in Fig. 7c–e. The colorized dots in (c) correspond to $T_{F(e)}$ (green) and $L_{F(e)}$ (blue) preferring neurons shown in Fig. 6. Here, these separate along the vertical axis as one should expect.

First, we tested the pooled dataset that included all *TLC* configurations in the response field analysis (Fig. 7c). There was no significant deviation (Wilcoxon signed rank test, $p = 0.79$), between the original data ($y$-axis, mean = 0.16 median = 0.10,

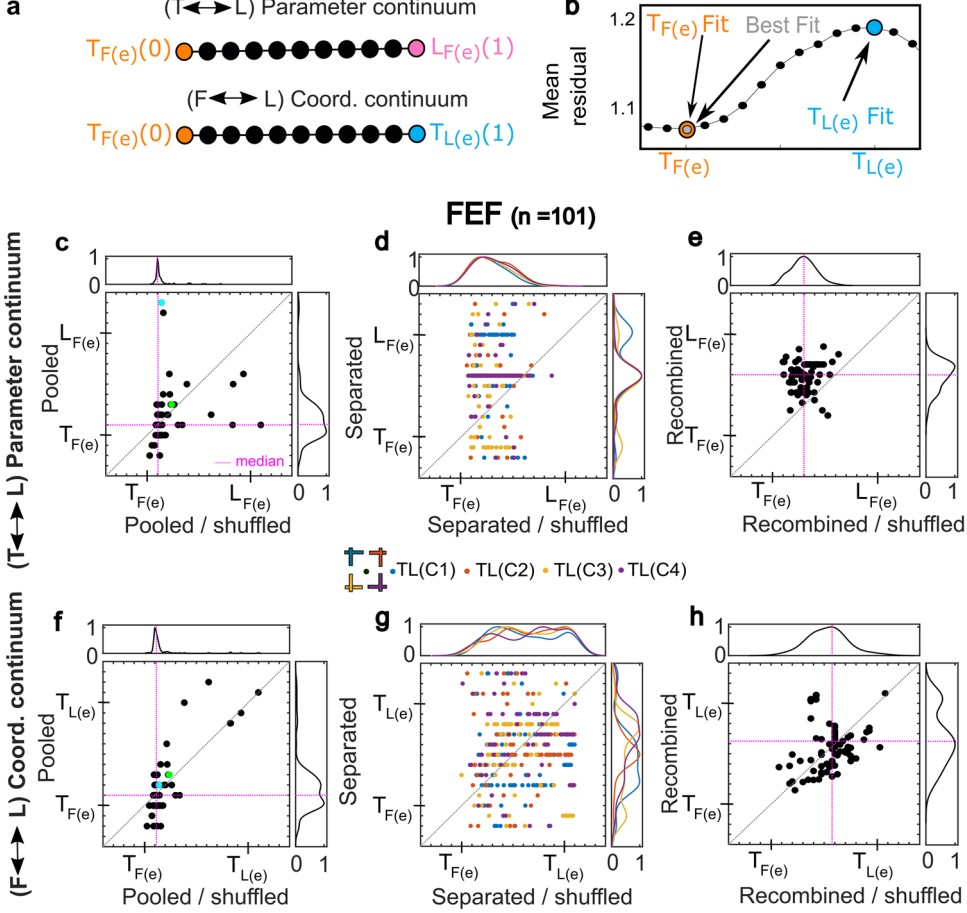

**Fig. 7 Intermediate spatial model analysis along the *T-L Parameter Continuum* and *F-L Coordinate Continuum* for FEF. a** Intermediate spatial models for both (1) Target-Landmark coding in the same fixation-centered coordinate system (the *T-L Parameter Continuum*) and (2) target coding between fixation-centered and landmark-centered coordinates (*F-L Coordinate Continuum*). Black circles show intermediate steps along the mathematical continuum between $T_{F(e)}$ (represented as 0) and one of the other two models (represented as 1.0). **b** Example distribution of mean residuals of fit along the *F-L Coordinate Continuum* for the neuron shown in Fig. 4, displaying minimal residuals (best fit) exactly at $T_{F(e)}$. **c–h** Complete Continuum analysis for all FEF neurons (see Supplementary Fig. 5 for a similar SEF analysis). In each panel, fits to experimental data are plotted as a function of the median best fit for their corresponding 100 shuffled control datasets (i.e., along x-axis each dot represents median of 100 shuffles) and their overall distributions are shown above/beside the plot. The magenta lines show the intersection of medians along x and y axes and the diagonal dotted lines show the line of equality between the pooled and shuffled control fits. To be comprehensive, we included neurons with and without spatial tuning in this analysis. **c–e** Swarm plot charts of the distributions of best fits along the *T-L Parameter* continuum. **c** Fits for response fields where data were pooled across different target-landmark configurations. **d** Configuration-dependent analysis. Each color represents a specific target-landmark configuration. **e** Recombined data (mean of 4 median fits/cell). **f–h** Similar analysis (as **c–e**) but for the *F-L Coordinate Continuum*. **f** Fits for pooled configuration data. **g** Configuration-dependent Fits. **h** Recombined data fits. Note that FEF/SEF neurons that were spatially tuned always showed a significantly shifted coding toward both $L_{F(e)}$ and $T_{L(e)}$ for at least one target-landmark configuration, whereas untuned neurons never showed significant shifts along either continuum for any *TLC*. The colorized dots in (**c**) and (**f**) correspond to $T_{F(e)}$ (green) and $L_{F(e)}$ (blue) preferring neurons shown in Fig. 6.

std = 0.22) and the shuffled control data (*x*-axis, mean = 0.12, median = 0.10, std = 0.10), i.e., between the median of the original data and a control median derived from the medians of the 100 shuffled datasets. However, the distribution spreads (shown above and beside the scatter plot) were significantly different (10,000 bootstraps, 95% confidence interval). Specifically, the original data fits (vertical distribution) were much broader than the control fits (horizontal distribution), suggesting more variable levels of target vs. landmark coding in the experimental versus control data. At the single cell level, 26.3% of the FEF neurons showed significant landmark coding relative to their own control data (95% bootstrapped confidence interval). SEF neurons showed similar, but more modest trends (Supplementary Fig. 5a), with broader distributions but only 13.6% of neurons exhibiting significant landmark coding relative to controls.

Next, we tested if any new information emerged when target-landmark configuration was accounted for. To do this, we separated the pooled response field data for each cell based on the four different target landmark configurations (Fig. 1b), yielding four sub-sets with equal amounts of data and nearly identical spatial distributions (Supplementary Fig. 6). We then repeated our fits on each dataset.

Figure 7d shows the resulting *T-L Parameter Continuum* fits to the separated response field data, plotted relative to their shuffled controls. Notably, all these scatters were significantly different spread relative to the shuffled control (10,000 bootstrap, 95% confidence interval). Statistical analysis showed that now 50.5% of FEF neurons displayed a significant difference from their control data for at least one configuration (95% bootstrapped confidence interval). At the level of individual target-landmark configurations, different fractions of data were significant

(*TLC 1-4*: 23.2%/32.3%/15.2%/15.2%, 95% bootstrapped confidence interval). A similar trend was observed for SEF neurons, with 31.8% of them being significant for at least one configuration (Supplementary Fig. 5b). Overall, this suggests that many FEF (and some SEF) neurons still showed landmark coding for some configurations when tested on individual *TLCs*.

Finally, to test for a systematic landmark influence at the population level, after separate configuration analysis, we recombined this data by averaging the four *TLC* fits for each cell (Fig. 7e). Visually, the data distribution was now much less variable, likely because both noise and real opposite target-landmark effects canceled out. However, at the population level, FEF now showed a significant upward shift along the *T-L Parameter Continuum* toward $L_{F(e)}$ (Wilcoxon signed rank test, $p = 4.25 \times 10^{-12}$; relative to controls), suggesting an overall shift toward coding the landmark that was not evident in the pooled data. In comparison, the shift in SEF only approached significance (Wilcoxon signed-rank test, $p = 0.051$) (Supplementary Fig. 5c). In addition, 51.52% of the individual FEF neurons (15.91% in SEF) showed a significant shift (95% bootstrapped confidence interval).

Collectively, these results agree with Fig. 6e but further suggest that the FEF visual response (in SEF to a lesser degree) multiplexes both target and landmark information at the single unit and population levels. They also suggest that some information was lost when we pooled data across configurations, likely because of the direction-specific effects.

**Intermediate target coding schemes: the *F-L* coordinate continuum**. Another key question is whether the visual landmarks influence the coordinate frame used to represent the target[30,31]. Relatively few FEF neurons showed such an influence when only cardinal models were compared (Fig. 6e). But here, we hypothesized that neurons might encode the target position along an intermediate reference frame between gaze-centered and landmark-centered coding. To test this, we performed fits along the *F-L Coordinate Continuum*. As before, we used our shuffled dataset as a control, initially pooled (Fig. 7f) across different landmark configurations, then separated (Fig. 7g) and recombined (Fig. 7h) for different landmark configurations.

Figure 7f contrasts the population distributions of the original (*y*-axis) versus the shuffled controls (*x*-axis) along the *F-L Coordinate Continuum* for pooled FEF response fields. In this case the two neurons from Fig. 6 (green and blue dots) cluster together as they should, because they both had the same coordinate system [$F_{(e)}$].

Again, the original distribution had a significantly larger spread (10,000 bootstraps, 95% confidence interval). Specifically, 66.7% of the FEF neurons showed a significant landmark influence relative to their corresponding control data (95% bootstrapped confidence interval). Once again, this influence was bidirectional, canceling out at the population level (Wilcoxon signed rank test, $p = 0.71$) between the original (mean = 0.17, median = 0.1, std = 0.25) and the control datasets (mean = 0.16, median = 0.10, std = 0.18). Similar, but more modest, trends were observed for the SEF (Supplementary Fig. 5d), with 56.8% neurons showing significant landmark coding.

We then repeated the same response field analysis separately for each of the four *TLC* configurations. The spread of datapoints was significantly different in both FEF (Fig. 7g) and SEF (Supplementary Fig. 5e) compared with the pooled data. However, these fits still exhibited a significantly wide distribution relative to the shuffled control, with 53.5% of FEF and 52.3% of SEF neurons displaying a significant difference for at least one configuration. At the level of individual target-landmark

configurations, different fractions of data were significant for both the FEF (*TLC-4*: 31.3%/37.4%/18.2%/15.2%) and SEF (*TLC1-4*: 27.3%/25.0%/27.3%/29.6%). When these fits were recombined (Fig. 7h), FEF showed a modest but significant shift toward $T_{L(e)}$ at the population level (Wilcoxon signed rank test, $p = 0.002$) but SEF showed no population shift (Wilcoxon signed rank test, $p = 0.8$, Supplementary Fig. 5f). Overall, 39.4% of the individual FEF neurons (27.3% in SEF) showed a significant systematic shift across all four landmark configurations (95% bootstrapped confidence interval).

Collectively, these results suggest that the FEF/SEF employs intermediate coordinate frames, with significant landmark influence in most cells. Most of these opposing influences on individual cells seemed to 'wash out' at the population level, until FEF neurons were fit separately for different TLCs and the fits recombined.

**Target-landmark integration: a cell-level coordinate transformation**. Finally, we asked if there might be some relationship between the landmark and target coding in our cell populations. One possibility is that the landmark codes we observed in our visual responses are just residual noise from the visual system and have no local influence on target coding. In this case, the *T-L Parameter Continuum* and *F-L Coordinate Continuum* from the last two sections should be independent. Alternatively, if landmark information had a local influence on target coding in prefrontal cortex, these two continua fits should interact. Specifically, if this interaction occurs within individual cells, one would expect response fields that code *both* landmarks and targets to also show the biggest shift toward landmark-centered target coding. In other words, *F-L Coordinate* scores should peak somewhere near the middle of the *T-L Parameter* distribution.

To test this, we plotted the best-fit scores of neurons along the *F-L Coordinate Continuum vs.* the *T-L Parameter Continuum* scores for both FEF and SEF (Fig. 8). For this analysis, we used the configuration-dependent dataset (from Fig. 7, middle column) to maximize the data spread and information content, but only included spatially tuned data to minimize noise. Both FEF (Fig. 8a) and SEF (Fig. 8b) show the same pattern: influence on target coding grows from near zero at the horizontal edges (at pure target or landmark coding) toward a central peak near the point where target and landmark coding are equal. At this peak *F-L Coordinate* scores range (vertically) from approximately equal target-landmark weighting on the coordinate system to purely landmark-centered target coding. Note that this distribution can be approximated by two curves (green and blue lines) that asymptote at the peak. These patterns are not a trivial result of our fitting algorithm, because they did not emerge when we plotted the shuffled control data in the same way (Supplementary Fig. 7).

From these results, we conclude that not only are landmark signals preserved in the frontal cortex visual responses, but they also interact locally with saccade target responses to influence their spatial coding scheme: specifically, shifting the coordinate frame for target coding from fixation-centered (egocentric) toward landmark-centered (allocentric) coding.

## Discussion

In summary, our analysis shows that visual landmark information is preserved in the visual target responses of prefrontal gaze structures and has an influence on saccade target coding. Eye-centered landmark coding was the second most common signal observed in FEF (after eye-centered target coding) and was multiplexed in most FEF/SEF cells. In addition, most FEF/SEF cells showed modest intermediate shifts toward Landmark-

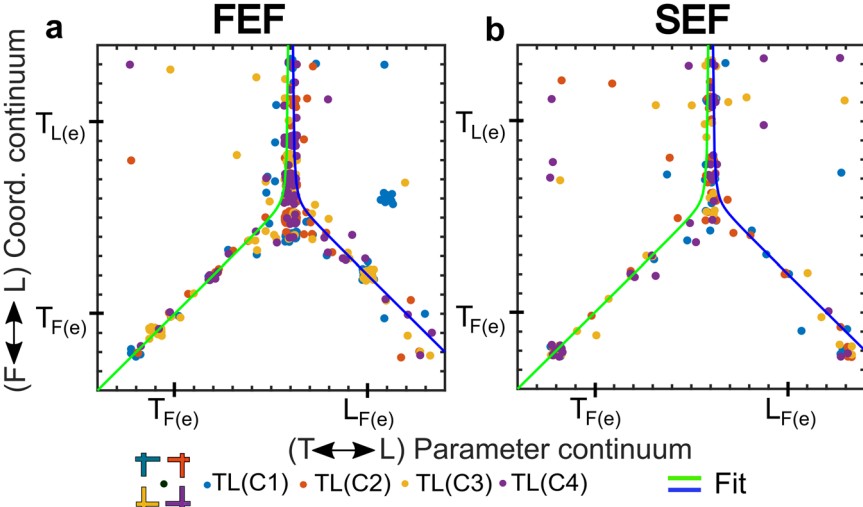

**Fig. 8 Target-landmark integration produces the fixation-landmark coordinate shift at a cellular level.** *F-L Coordinate Continuum* scores are plotted as a function of *T-L Parameter Continuum* Scores for both **a** FEF and **b** SEF. Configuration-dependent fits are used to provide the richest dataset with the broadest distribution. Note that in both cases, the coordinate shift along the vertical axis rises linearly toward center from both fixation-centered extremes [$T_{F(e)}$ and $L_{F(e)}$] along the horizontal axis, then rises rapidly toward a peak near Landmark-centered coding. In other words, cells that coded both targets and landmarks also showed the biggest coordinate frame shift. The green and blue lines show separate fits (see "Methods" for details) to each side of the data illustrating a hypothetical asymptote at the peak. These patterns were not observed in fits to the shuffled control data (Supplementary Fig. 7).

centered target coding. In both cases, the effect was stronger when target-landmark configurations were separated and then recombined. And finally, cells that showed both target and landmark coding also showed the most landmark-centered target coding.

FEF visual responses are generally associated with encoding potential saccade targets in eye-centered coordinates[31,37,49–53]. This remains the case for FEF and SEF in the presence of a complex visual background[30,31]. This was confirmed in our initial analysis (Fig. 5), where we pooled that data both across cells and target-landmark configurations, and only analyzed the visual response using 'cardinal models'.

However, it now appears that these conventions obscured several important landmark effects in the data. First, even though we did not explicitly train monkeys to attend to the landmark (and it only produced weak visual response; Supplementary Fig. 2), landmark coding dominated 30% of the FEF target responses. Further, FEF responses showed significant shifts along our *Target-Landmark Parameter Continuum*, both at the population level and in many individual cells. This suggests that many FEF cells encoded both targets and landmarks (target-landmark multiplexing). These findings demonstrate persistence of a signal normally associated with ventral stream vision in prefrontal cortex[54–56], but see the paper by Rao and colleagues[57]. Consistent with this, when we trained a deep neural net to perform the same task, both target and landmark coding appeared in the intermediate layers, and this was then integrated into an eye-centered gaze command at the output layer, similar to FEF motor responses[14].

In comparison, relatively few SEF cells preferentially coded landmark location and the SEF population did not show a significant shift along the T-L Parameter Continuum relative to controls. This might be a statistical effect, because fewer SEF cells were spatially tuned compared with FEF, effectively reducing the N. Alternatively, if the negative SEF result is real, it might be because the SEF is anatomically further removed than FEF from the visual input[58–61] and is also driven more by internal signals[62–65].

Finally, it is possible (perhaps likely) that these signals were enhanced by long-term exposure to the landmarks in our task. If

so, this must have occurred through some endogenous process since we only trained our animals to look at the saccade target. The same supposition would thus suggest that animals would also develop prefrontal responses to task-relevant landmark cues that predominate in their natural environment.

Several studies have reported that visual distractors can modify prefrontal visual responses to saccade targets[48,50,66–68], and other studies have shown that prefrontal saccade responses can be modulated by a target position within an object[69,70]. However, to our knowledge, this is the first study that has systematically studied the influence of a stationary, independent, and reward-irrelevant landmark on the coordinate frames of frontal cortex *visual* signals.

In our dataset, the landmark had a significant but variable influence on most individual FEF/SEF cell responses, showing both attracting and repelling properties. Specifically, a neuron's response was dictated by both the location of the target relative to Fixation [$T_{F(e)}$] and the landmark [$T_{L(e)}$] in eye-fixed coordinates, resulting in a significantly larger spread of data along the continuum from eye-centered toward landmark-centered coding (relative to controls) in both FEF (Fig. 7f) and SEF (Supplementary Fig. 5d). This does not mean that response fields shifted *toward* or away from the landmark, but rather the landmark had some influence on the coordinate frame. In extreme cases, some response fields appeared to be fixed relative to the landmark (Fig. 7, *bottom row*). In this case, the response field should shift along with a shifting landmark. However, our landmark- and fixation-centered coordinates were mainly dissociated by trial-to-trial variations in eye torsion in our paradigm (Fig. 2), which did not create systematic response field shifts. Other studies have shown intermediate (e.g., eye-head) egocentric frames[2,8,46], but to our knowledge, this is the first time an intermediate ego/allo-centric code has been demonstrated in the visuomotor system.

The ventral visual stream and hippocampus are replete with configurational information, coding features relative to other features[71,72] and objects relative to other objects[73,74], respectively. This configurational coding may be its key distinction from the dorsal stream, which is typically considered to code point locations in absolute (egocentric) coordinates[15,54]. A relevant

exception is that some fMRI studies suggest that certain fronto-parietal areas appear to code left versus right saccade directions relative to a landmark[15,19,55,75].

Configurational information also influenced our results. When we fit intermediate codes to our original pooled dataset, they showed no significant shift at the population level, possibly because opposing configuration dependencies 'washed out' (Fig. 7c, f). Consistent with this, when we separated the data by target-landmark configuration, the parameter and coordinate continua fits showed significantly broader spreads (Fig. 7d, g). And when these data were recombined, they showed significant shifts toward $L_{F(e)}$ and $T_{L(e)}$ coding (Fig. 7e, h). These factors suggest a configuration-specific influence, either in the input to, or within frontal cortex (see next section).

The cellular mechanisms for integration of egocentric and allocentric information are well described in hippocampal systems for memory and navigation[76,77], but to our knowledge, this has never been described for visual responses in the sensorimotor system. Our final analysis (Fig. 8) strongly suggests that target and landmark information interact at the cellular level in prefrontal cortex. Specifically, FEF and SEF neurons that multiplexed both target and landmark signals also showed the strongest landmark influence on saccade target coding. Presumably, these interactions occur between the weak, static response to the landmark (which remained on during the presentation of target) and input specific to the target. This suggests (1) that target-landmark information is not fully integrated until it reaches frontal cortex and (2) shows that the mechanism for this integration involves calculations within individual cells.

How are target/landmark signals relayed to frontal cortex? Single-point target information is propagated throughout the gaze system[58,78–80], but it is not clear how landmark information reaches the frontal cortex. Ultimately, the occipital cortex possesses the necessary machinery to process landmarks and projects to both parietal cortex and temporal cortex, associated with egocentric and allocentric coding respectively[2,16,17]. Likewise, extrastriate visual cortex contributes to the allocentric codes observed in hippocampus[81–84]. However, parietal cortex also shows landmark-influenced saccade signals[15,25] and projects to both SEF and FEF[48,85]. A recent study has suggested that the hippocampus and the gaze system are more closely linked[86] than previously thought, but it is not clear if the ventral visual stream/hippocampal complex has a direct influence on prefrontal landmark codes, or if this might occur via parietal cortex[87–89].

Presumably the visual signals described here are ultimately responsible for the landmark influence that we previously observed in FEF/SEF memory and motor responses[30,31]. However, given the sluggish nature of those memory/motor responses, it is likely that landmark-related signals are further preserved and refined by distributed interconnections between the prefrontal memory/oculomotor systems[65,90,91].

Why does the sensorimotor system need visual landmarks? It is thought that internal copies of 3D eye and head orientation are used both in sensorimotor transformations[39,92] and perception[5,34]. However, such signals are inherently noisy[7,10,29,93,94], and this can directly reduce the precision of action. This is particularly relevant in natural, head-unrestrained conditions, where gaze fixations include variable initial eye and head orientation, including relatively large variations in eye torsion (Fig. 2a). Visual landmarks can help compensate for the noise in this system[6,8,15,72,95], as shown previously in the same animals and behavioral paradigm used here[7]. Possibly, the push-pull effect of the opposing landmark influences that we observed at the single cell level (Fig. 8) could contribute to this stabilizing effect.

In our experiment we utilized a single, simple landmark so that we could clearly quantify its influence, but the real world is generally replete with potential landmarks. It is likely that landmarks have an even stronger influence in natural settings. Indeed, an extensive background shift has more influence on visual responses than a slight landmark shift[25,96], and conversely, should have a more stabilizing influence on vision and behavior when stationary. Further, in real-world conditions, not all landmarks are equal: landmarks differentially influence behavior based on salience, distance, reliability, task-relevance and prior experience[3,10,11,29,96]. Our landmark task approximates the case where the visual response is dominated by a single nearby, salient, and stable landmark.

This study provides several fundamental insights into the way the prefrontal cortex encodes visual information, and how it processes this information for goal-directed action. Specifically, we found that prefrontal visual responses multiplex both saccade target and landmark information, that these signals are configuration-dependent, and that they interact to produce a shift toward landmark-centered coding of gaze targets. Taken together with previous behavioral, neuroimaging and neurophysiological results[7,10,30,31,72], these results suggest that prefrontal cortex is involved in the use of visual landmarks to stabilize gaze goals in the presence of noisy internal signals. We expect this is also the case within the frontal mechanisms for other goal-directed behaviors, such as reaching.

## Methods

Although experimental details have been published previously[30,31] but are also detailed here, along with descriptions of the new configuration-dependent and intermediate ego/allocentric frame analysis methods employed in this study.

**Surgical procedures and recordings of 3D gaze, eye, and head.** All experimental procedures were approved by the York University Animal Care Committee and were in accordance with the guidelines of the Canadian Council on Animal Care on the use of laboratory animals. The neural data used in this study were collected from two female *Macaca mulatta* monkeys (Monkey V and Monkey L, both animals were aged 10). Surgeries were performed to plant the chambers and the search coils[33]. Both animals were implanted with 2D and 3D search coils. Both search coils had a diameter of 5 mm and were implanted in the sclera of the respective animal's left eye. The recording chambers for both animals were implanted centered at 25 mm anterior and 19 mm lateral for FEF and 25 mm anterior and 0 mm lateral for SEF. Underneath each chamber was a craniotomy of 19 mm diameter to allow access to the right FEF and the right SEF. During the experiment, the animals were placed in a custom-made primate chair modified to allow free head movements.

In addition, the monkey was suited with a vest connected to the primate chair to restrict it from rotating around in the chair. Furthermore, two orthogonal coils were mounted on the head of the monkeys during the experiment. The animal was then placed in the setup, which was equipped with three orthogonal magnetic fields. These fields induced a current in each coil. The amount of current induced by each of the fields is proportional to the coil area parallel to this field. Thus, allowing to derive the orientation of each coil in relation to the magnetic fields and in turn, the orientations, velocities, and accelerations of the eye and the head of the animal[33].

**Behavioral paradigm.** Using a back projector (NEC UM330X), the visual stimuli were presented on a flat screen located 80 cm in front of the animal. The animals were trained on a memory-

guided cue-conflict saccade task, where the monkey had to perform a saccade to a remembered target relative to an allocentric landmark (two intersecting lines) that shifted during the memory delay after a mask presentation (Supplementary Fig. 1).

Each trial started with the monkey fixating a red dot located centrally on the screen for 500 ms in the presence of the landmark. Then a white dot serving as visual target was briefly flashed for 100 ms in one of four oblique positions relative to the landmark vertex. Within the context of this paper, each of these target-landmark combinations will be called target landmark configurations [*TLC1 (45°), TLC2 (135°), TLC3 (−135°),* and *TLC4 (−45°)*]. For example, *TLC1* refers to the Target-landmark configuration where the landmark was present at a 45° angle, 11° away from the target. It is the visual response in FEF/SEF neurons to these target-landmark configurations that was analyzed in the current study.

The above events were followed by other events that were described in our previous publications[30,31]. Following a delay of 300 ms, a grid-like mask was displayed for 200 ms to occlude visual traces of the landmark and the target. After the offset of the mask, the landmark reappeared either shifted (90% of cases) by 8° in one of eight equally spaced radial directions or not shifted (10% of cases). Following a random delay between 200–600 ms, the fixation point disappeared acting as a go signal for the animal to initiate a saccade. If the gaze of the monkey landed anywhere in an 8–12° radius around the original target position, the monkey received a droplet of water as reward. This large reward window ensured the monkey was not biased towards either the original target location or the virtually shifted target location fixed to the shifted landmark. Note that all angles mentioned in this section were assumed to be linear. This means an 8° shift in the center of the screen stretches over the same distance on the screen as an 8° shift at the outskirts of the screen.

**Behavioral recordings, electrophysiological recordings, response field mapping, and data inclusion**. During the experiment, 3D eye and head orientations in space were recorded at a sampling rate of 1 kHz using the implanted and head-mounted search coils, respectively. These (as well as target and landmark locations) were recorded, analyzed offline, and then used to compute the spatial coordinates for various model fits, such as [$T_{F(e)}$, $T_{L(e)}$, $L_{F(e)}$] and various others[30,31]. Note that in the head-unrestrained range of gaze (where eye torsion is variable and the non-commutativity of rotations becomes prominent), linear approximations can produce large errors[5,97]. So, for example, $T_{F(e)}$ was computed by rotating the eye-target vector by the inverse of initial 3D eye orientation[98].

The neuronal activity in the FEF and SEF was recorded in parallel with tungsten microelectrodes (0.2–2.0 mΩ, FHC Inc.) using the 64 channel Plexon MAP system. To lower the electrodes, the Narishige MO-90 hydraulic micromanipulator was used. The recording sites of the FEF and the SEF were confirmed by using a low-threshold (50 µA) electrical microstimulation while the head was restrained[99]. In each session two electrodes were used (one was lowered in FEF and other was lowered in SEF). Neurons were mostly searched for while the animal was head-unrestrained scanning its environment. When a reliably spiking neuron was found, the experiment was started. After an initial sampling period for the response field's dimensions, we presented targets (randomly one-by-one) in a $4 \times 4$ to $7 \times 7$ array (each 5–10° apart from each other) spanning 30–80° across horizontal and vertical dimensions. We aimed to record approximately 10 trials/target, so the bigger the response field (and thus the more targets), the more the number of recorded trials was required and vice versa. We mapped the entire

response field because our analysis method (see below) is most sensitive to 'slopes' rather than peaks and valleys[40].

For analysis of the visual activity, a fixed 100-ms sampling window was chosen, ranging from 80–180 ms after the target onset. Only neurons that showed significant activation in the sampling window were included in the analysis. Furthermore, trials in which the animals did not successfully fixate on the home position were excluded. We recorded a total of 312 (140 from Monkey V and 172 from Monkey L) neurons in FEF and 256 (102 from Monkey V and 154 from Monkey L) neurons in SEF. Monkey V contributed 50 and 28 visual neurons in FEF and SEF, respectively, of which 39 were spatially tuned in FEF and 22 were spatially tuned in SEF. Monkey L contributed 104 and 31 visual neurons in FEF and SEF, respectively, of which 62 were spatially tuned in FEF and 21 were spatially tuned in SEF. The percentage of modulated and spatially tuned neurons in both areas is comparable to the literature[48,78,99–101].

**Fitting neuronal response fields against spatial models**. Each of the models tested in this study was derived from laboratory measurements and behavioral data. For example, each $T_{F(e)}$ (Target-relative-to-fixation in eye coordinates) position was computed by calculating the vector from the eye to the target in space and then rotating this by the inverse of 3D eye-in-space orientation quaternion at the fixation viewing time[102]. Likewise, $T_{L(e)}$ (Target-relative-to-landmark in eye coordinates) was derived by calculating the eye-target vector relative to the eye-landmark vector in space and then rotating this by the inverse of the 3D-eye position quaternion at fixation $Q_F$ for the entire series of trials (1).

$$T_{F(e),i} = Q_{F_i}^{-1} T_i \quad T_{L(e),i} = Q_{F_i}^{-1}\left(T_{F,i} - L_{F,i}\right) \tag{1}$$

For our method, to differentiate between such spatial models, they must be spatially separable, and this must vary across trials. This variability is ensured by the stimulus design (e.g., random fixation position) and the animal's natural behavior. Further, opposed to decoding approaches which typically test the set of parameters implicitly coded in population neuronal activity[103,104], our technique directly tests which underlying spatial model best explains variation in the neuronal activity. The response fields of neurons (A) were fitted against the different spatial models [for example $T_{F(e)}$ and $T_{L(e)}$] using a non-parametric fit with a Gaussian kernel in conjunction with Euclidian distance (d) as shown in Eqs. (2) and (3).

$$d_{TF(e),i,j} = \sqrt{\left(T_{F(e),i} - T_{F(e),j}\right)^2} \quad d_{TL(e),i,j} = \sqrt{\left(T_{L(e),i} - T_{L(e),j}\right)^2} \tag{2}$$

$$A_{fit}\left(T_{F(e),i}\right) = \sum_{j \neq i}^{n} A\left(T_{F(e),j}\right) e^{-\left(\frac{d_{TF,ij}}{KW}\right)^2} \quad A_{fit}\left(T_{L(e),i}\right) = \sum_{j \neq i}^{n} A(T_{L(e)j}) e^{-\left(\frac{d_{TL,ij}}{KW}\right)^2}$$
$$\tag{3}$$

Our fitting method determines the spatial coherence of the response field activity in different reference frames by employing non-parametric regression on neural data, and then calculating how good the regression surface was in predicting the unfitted data using the Predictive Sum-of-Squares (PRESS) statistic. Thus, the PRESS statistic allowed us to to quantify the quality of the fit ($A_{fit}$). The coordinate frame yielding the least PRESS statistic (residuals) was deemed to be the intrinsic reference frame. The residuals in different frames were tested for significance using the Brown–Forsythe test.

These residuals were calculated for each trial by fitting the response field by subtracting the data from the left-out trial and then comparing the activity predicted by the fit for the spatial properties present in the trial and the actual activity measured

during the trial. Afterward, these residuals were squared and averaged across each trial to derive PRESS value for a given fit. The bandwidth of the Gaussian kernel (KW) was determined for each neuron individually to match the response field's size, shape, and contour[40]. This was done by calculating the PRESS statistic for each spatial model for all bandwidths between 1 and 15. Then the bandwidth yielding the lowest residuals was deemed as the best fit or spatial model. A schematic of this is displayed in Fig. 4b. Put simply, neural data plotted in the correct reference frame/spatial model would lead to least residuals, e.g., a target-fixed response field would fit best in target-fixed coordinates, whereas landmark-centered coordinates would yield higher residuals. Note: Once the optimal kernel width was determined for a neuron, the same kernel width was used for each reference frame and each target-landmark configuration, as well as the pooled condition.

The method described above was used in our previous studies[30,31], but in that case we only tested the visual response fields at the population level, we pooled across target-landmark configurations, and only tested 'cardinal' models. We found that $T_{F(e)}$ model yielded lower residuals than Target-in-space, $T_{(s)}$, Target-in-head, $T_{(h)}$, Landmark-in-space, $L_{(s)}$; Landmark-relative-to-fixation in eye coordinates, $L_{F(e)}$; and Target-relative-to-landmark in eye coordinates, $T_{L(e)}$]. Here, we repeated the same analysis and obtained the same result (Fig. 5), but also tested individual neurons (Fig. 6), intermediate frames of reference, and separate target-landmark configurations (Figs. 7, 8), as described below.

**Pooled vs. separate analysis.** In our initial analysis, data for each neuron were pooled across trials and all four target-landmark configurations, but it can be argued that these configurations might have different (even opposite) influences that might cancel out. For the separate condition, trials were grouped with respect to the specific $TLC$ (Fig. 1b), i.e., the response fields were fit with the neural data from the trials only corresponding to a landmark in a specific direction (also referred to as direction-dependent analysis). Thus, resulting in four coding preferences/conditions for each neuron (one for each $TLC$). Since in this pipeline all $TLCs$ are viewed individually, the effects of the landmarks will not cancel out.

Note that in this case, the target-landmark vector was fixed, but variations in initial eye orientation caused this to vary relative to the retina, thus separating $T_{F(e)}$ and $T_{L(e)}$ (Fig. 2). Without this dissociation, it would not be possible to distinguish a shift in the coordinate systems vs. a shift in response field activity within a fixed coordinate system, at least in our segregated datasets. This is another reason why our behavioral recordings were important: variations in eye orientation are larger and more variable without head-restraint[33], and 3D eye recordings were needed to account for this.

**Intermediate spatial models.** Our previous results[30,31,37,41,105] suggested that neuronal response fields do not always exactly fit the canonical spatial models like $T_{F(e)}$, but instead might best be described by intermediate models between the canonical ones (Fig. 7a). However, in those studies we only looked at the intermediate models within or between egocentric frames of reference, using linear interpolation. In this study, we investigated spatial models that exist between the egocentric and allocentric frames (Eqs. 4, 5, 6). This makes it impossible to calculate intermediate spatial models by interpolation. So instead, we incorporated a weighting factor (w) into our algorithm (Eqs. 5, 6).

For example, to calculate Target positions along the *Fixation-Landmark Coordinate Continuum*, we first calculated the

distances (d) between trials used in the non-parametric fit used for response field fitting by calculating the Euclidian distance (5) between four-dimensional vectors representing the trials.

$$T_{F(e),i} = \begin{pmatrix} x_{TF(e),i} \\ Y_{TF(e),i} \end{pmatrix} \quad T_{L(e),i} = \begin{pmatrix} x_{TL(e),i} \\ Y_{TL(e),i} \end{pmatrix} \tag{4}$$

$$d_{w,i,j} = \sqrt{\left( \begin{pmatrix} wT_{F(e),i} \\ (1-w)T_{L(e)i} \end{pmatrix} - \begin{pmatrix} wT_{F(e),j} \\ (1-w)T_{L(e)j} \end{pmatrix} \right)^2} \tag{5}$$

$$A_{fit,w}\left(T_{F(e),i}, T_{L(e),i}\right) = \sum_{j \neq i}^{n} A(T_{F(e),j}, T_{L(e),j}) e^{-\left(\frac{d_{w,i,j}}{KW}\right)^2} \tag{6}$$

$T_{L(e)}$ gives the first two elements of these four-dimensional vectors, and the last two elements are given by $T_{F(e)}$. The continuum is derived by weighting the first two elements against the last two elements during the Euclidian distance calculation. Thus, the continuum ranges from $T_{F(e)}$ [weight of $T_{F(e)} = 1$, weight of $T_{(Fe)} = 0$] to $T_{L(e)}$ [weight of $T_{L(e)} = 0$, weight of $T_{L(e)} = 1$] with nine steps in between (Fig. 7a). An example of response field fitting for a continuum between two reference frames [$T_{F(e)}$ and $T_{L(e)}$] is displayed in Fig. 7b. The residuals for each fit are displayed along the continuum ranging from $T_{F(e)}$ (0) to $T_{L(e)}$ (1), with the best fitting step exactly at 0, i.e., $T_{F(e)}$ (gray dot). A similar algorithm was used to compute intermediate points along the *Target-Landmark Parameter Continuum* in the same [$F(e)$] coordinate system.

**Test for spatial tuning.** In order to determine what spatial frame best describes a response field, one must first confirm that the neuron has a spatially tuned response field. This does not imply that the spatially untuned neurons do not contribute to the implicit population codes[106–110]. For example, some studies have reported that decoding information from neurons works better when both tuned and untuned neurons are included in the population[103,108,109,111]. Note that these two approaches are complementary: decoding describes information that can be extracted from an unknown spatial code, whereas our technique attempts to reveal the spatial codes that neurons themselves are using. To test for spatial tuning, the firing rate data were shuffled over the target position data obtained from the best-fitting model[37,40]. The mean PRESS residual distribution (PRESS_random) of the 100 randomly generated response fields was then statistically compared with the mean PRESS residual (PRESS_best-fit) distribution of the best-fit model (unshuffled, original data). If the best-fit mean PRESS fell outside of the 95% confidence interval of the distribution of the shuffled mean PRESS, then the neuron's activity was deemed spatially selective. We defined an index (Coherence Index, CI, Eq. 7) for spatial tuning, which was calculated as[38]:

$$CI = 1 - \left( \frac{PRES_{best-fit}}{PRSS_{random}} \right) \tag{7}$$

If the PRESS_best-fit was like PRESS_random then the CI would be roughly 0, whereas if the best-fit model is a perfect fit (i.e., PRESS_best-fit = 0), then the CI would be 1. Unless otherwise stated, we only included those neurons in our analysis that showed significant spatial tuning.

**Test against randomized/shuffled control.** To ensure that the coding preferences along our spatial continua were not just a result of random noise fitting, a test on each individual neuron coding relative to a randomized control dataset was conducted. This control dataset was created by using the Matlab RandPem Function to randomly shuffle the landmark position information

relative to the target-neural response pairing for each trial. For the pooled-configuration analysis, this randomization was repeated 100 times for each cell, and continuum fits were made to the response field for each shuffled dataset, thus creating a distribution of 100 fits. These 100 fits were then used as a control against which the data was compared.

On population level, this comparison was done by testing the actual data against the shuffled data with the Wilcoxon rank sum test. For individual cells, we tested if the bootstrapped (100 straps) 5% confidence intervals of the difference between the continuum step of the actual data fit and the continuum step of the corresponding shuffle fit distributions contained zero. If zero is not contained in each 5% confidence interval, the neuron is considered to have a significant deviation between the actual and shuffled data.

In the case of the configuration-dependent analysis, the same procedures were followed. However, before generating the shuffled fits, the data were separated into four datasets according to their original (unshuffled) landmark configuration.

**Fits: F-L coordinate continuum vs. T-L parameter continuum**. The asymptotes to the F-L coordinate vs. T-L parameter continuum were fit using the following function (free parameters were chosen based on the shape of the data):

$$\frac{1}{a(x-b)^2} + cx = d \qquad (8)$$

Where, $a = 100$ (for smoothness), $b = 0.6$ (asymptote point), $c = 1/-1$ (left side $= 1$, right side $= -1$), and $d = 0/1.2$ (left side $= 0$, right side $= 1.2$, these are given by the y intersect). This can be read from the shape of the points. We decided on this construction approach because the optimal parameters drop down from the figure rather quickly.

**Statistics and reproducibility**. All statistical analyses were performed using MATLAB R2022a. We assumed a significance level of $p < 0.05$ for all statistical tests. Two animals (Monkey V and Monkey L) were used for this study and across both animals, a total of 312 and 256 neurons were recorded in FEF and SEF, respectively. Using rigorous statistical analysis (CI as defined above) for spatial tuning 101 FEF and 43 SEF neurons with visual responses were brought forward for the analysis reported in this paper. Note: the motor responses were not analyzed in this study.

**Reporting summary**. Further information on research design is available in the Nature Portfolio Reporting Summary linked to this article.

## Data availability

The preprocessed neurophysiological dataset and the numerical source data for main and supplementary figures can be downloaded here: https://github.com/bhav2501/Landmark_Paper_data.

## Code availability

The custom codes used during the current study are available from the corresponding author on reasonable request.

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

## Acknowledgements

This project was supported by a Canadian Institutes for Health Research (CIHR) Grant (Grant # MOP-130444) and the Vision: Science to Applications (VISTA) Program, which is supported in part by the Canada first Research Excellence Fund, by Deutsche Forschungsgemeinschaft (DFG: IRTG-1901, RU-1847, and CRC/TRR-135, project number 222641018) and the HMWK cluster project: The Adaptive Mind. V.B., X.Y., and H.W. were supported by CIHR and VISTA. J.D.C. is supported by the Canada Research Chair Program.

## Author contributions

A.S. performed the data analysis, contributed to interpretation and writing. V.B. did the experiments, helped with surgeries, contributed to data analysis, interpretation of results, and writing and editing of the manuscript. X.Y. helped in the technical aspects of recording the data. H.W. performed the surgeries and helped in neural recordings. F.B. contributed to result interpretation, writing, and editing of the manuscript. J.D.C. conceived the study and contributed to data analysis, writing, and editing of the manuscript.

## Competing interests

The authors declare no competing interests.
