## [Peer Review File · Communications Biology]

Reviewers' comments:

Reviewer #1 (Remarks to the Author):

This paper investigates the spatial frame of reference (FOR) used by the prefrontal cortex to encode visual stimuli locations in the presence of a visual landmark. Specifically, using a modified memory guided saccade task, the authors showed that the visual response of the future saccadic target was encoded in an intermediate FOR between a gaze and a landmark centered FORs. Those results are interesting and in line with findings showing that humans do use visual landmarks to support goal directed behavior. This paper shows a potential neuronal implementation supporting the use of landmarks in the saccadic system.

The authors ability to measure 3d eye and head movements in a head free configuration was critical to be able to distinguish between a gaze and a landmark centered encoding. The authors fitted the visual responses to those 2 FORs and to a range of combination of weighted version of the two. To measure the landmarks effects the authors rightly aggregated together trials with the same Target-Landmark spatial configuration. The results show that the visual responses are better explained by a weighted combination of the gaze and a landmark centered FORs. The use of more landmark centered spatial coding by the frontal cortex was interpreted by the authors as the brain's way to compensate for the variations of the eye and head position in space when the saccadic target was first shown.

The findings are interesting, but the writing style was hard to follow. Bellow you can find my detailed comments.

Major comments

1) The paper needs general improvement in the language used to explain the results: for example, in page 2 in the introduction, line 20, ".....nearby may modulate these codes..." the authors do not explain what they mean by modulate. Another example is when explaining the task: page 5, line 3, "...This cross could appear in one..." it seems that the landmark appears after the target is flashed. The authors clearly indicated that the landmark is already there when the target is flashed, but the sentence as it is made me confused. Example3, page 7, line 8, "...variable and arbitrary behavior...." Please explain what would be an example of arbitrary behavior in the current context.

2) My understanding is that ones a neuron is isolated, the authors made an initial mapping of the visual receptive field location to estimate it size. Only then, the authors proceeded with the task. The proportion of the visually responsive neurons to the spatially tuned ones is low (for FEF, 102 out of 312; for SEF, only 43 out of 256). The authors suggested that even the non-tuned neurons can potentially contribute to the code. The authors did not speculate how this can be the case in the current study.

3) In the discussion, line 8, the authors mentioned that the landmark was behaviorally irrelevant by task design. Indeed, the monkeys were rewarded for making a saccade towards the remembered location and the error window was big enough to include the original location (in 10% of trials) and the shifted location when the landmark shifted (in 90% of trials). Nevertheless, the authors found that the visual response actually shifted towards the landmark, even before the go signal was given. How the authors can rule out that the monkey's strategy was to encode the target location with respect to the landmark rather than the initial fixation? Ultimately, the landmark was a prominent visual cue on the screen and the target was shown at a fix location (one location out of 4 possibilities) with respect to it. In that case, the landmark might have been used an anchor and might have been behaviorally relevant as the monkey might have used it to solve the task.

4) The authors did not show how the behavior might be related to the FOR used to encode the saccadic target. A simple way to do so is to correlate the FOR used by the neurons and the FOR used to determine the end point of the saccade.

Minor comments

- 1) The authors did not mention how many cells each monkey contributed with to the data set.
- 2) The authors did not mention how many electrodes were used in each session.
- 3) Figure 5 top right plot, n=101 must be n=102.
- 4) Page 16 (Discussion), line 13, variable must be variable.
- 5) Page 18, line 19, '...the factors...' must be '...these factors....'.
- 6) Page 21, line 8, '...then used as to compute...' must be '...then used to compute...'.
- 7) Page 21, line 10, it seems that the authors cited a paper, but the editor did not find the corresponding reference.
- 8) Page 22, line 6, it seems that the authors cited a paper, but the editor did not find the corresponding reference.
- 9) Page 24, line 9, '.....in eye orientationare....' must be '.....in eye orientation are....'.
- 10) Page 14, line 3, '.....to test these results and the single.....' must be '.....to test these results at the single.....'.

Nabil Daddaoua

Reviewer #3 (Remarks to the Author):

In their study, the authors investigate the impact of visual landmarks on the neural encoding of visual saccade cues in the frontal lobe areas FEF and SEF. For this, they carried out an experiment in which rhesus monkeys made saccadic eye movements from a fixation point to the remembered location of a visual target. During the brief presentation of the visual target two intersecting lines were shown on the screen as visual landmark; the visual target briefly flashed in one of the four oblique directions relative to the crossing point of the lines, leading to four target-landmark configurations (TLCs). The cue presentation was followed by a memory period and a mask stimulus. Later the visual landmark reappeared, in some trials randomly shifted in one of the eight radial directions relative to its original position. Monkeys had to saccade to the remembered location of the visual target while their gaze was controlled with unrestrained head movements. A model was fitted to the visual response of the neurons recorded in FEF and SEF as a function of target location characterized as a 4D vector consisting of its weighted location relative to fixation and relative to the landmark to quantify the relative impact of either frame of reference. The main result is that when the model was fitted to different TLCs separately, a mixed representation of target relative to fixation and relative to landmark resulted in the best fit, i.e. the landmark had a partial impact on cue encoding. Instead, when the model was fitted to pooled data across all target-landmark configurations, target-to-fixation returned the best fit, i.e. landmark had seemingly no impact. The authors concluded that landmark influence depends on the relative positioning of the target and landmark, but not in a geometrically systematic way.

If I understand correctly, data of this study was published before in REFs 6, 7 (and 27? or just the same paradigm but different datasets were used before – please clarify). Previous studies had a focus on other time periods in the trial. The current study revisits the question of landmark-relative visual encoding, asking if a TLC-specific analysis of visual cue responses yields additional insights compared to the TLC-pooled analyses; in the latter, a gaze-centered (“target-to-fixation”) encoding was observed before.

The research is potentially interesting to researchers of visual sensory processing and higher cognitive functions such as spatial cognition and cognitive motor functions. The research topic of allocentric

encoding has never lost its timeliness, since fundamental aspect of this important form of spatial cognition remain unanswered until today. The current study has the potential to add detail to previous findings, but at this stage, the somewhat curious observations presented seem not to provide the major step forward in our deeper understanding.

Main comments:

- To say it in the words of the authors: "It is important to note that this TF-TL shift [i.e. the evidence for partial landmark-relative encoding] does not mean response fields shifted toward the landmark, which might indicate a simple shift in the perceived target direction. The small response field shifts that we observed (e.g., Figure 4 c3) were highly variable and did not follow any clear pattern across neurons." Having said this, the authors should provide convincing evidence that their model is not just tracking noise. They try so by comparing against shuffled surrogate data, but it is not fully clear to me, how with the $N=100$ randomizations they exactly to the statistically testing for significance of the observed result against the surrogate. More detail will be helpful to convince. At this stage the conclusions seem unsatisfying, since they suggest that in a previous publication a non-justified pooling was applied, washing out potentially interesting effects; but now that a more appropriate approach is taken, it is still not really clear what the impact of the landmark is all about. I wonder if this the right stage of the study to publish it.

-- Side note: The summarizing statement at the beginning of discussion "the landmarks caused a shift in the reference frame for coding the target toward the landmark (TL)" seems to contradict the above statement that results do NOT mean a RF shift toward the landmark. Please phrase such that confusion is avoided.

- As explained in the methods, "The bandwidth of the Gaussian kernel (KW) was determined for each neuron individually to match the response field's size, shape, and contour. This was done by calculating the PRESS statistic for each spatial model for all bandwidths between 1 and 15. Then the bandwidth yielding the lowest residuals was deemed as the best fit or spatial model." It is not clear whether the best models have different (KW) for different target-landmark configurations and pooled conditions? If that is the case, how did authors rule out the possibility that the reason why ($0 < W < 1$) returns the best fit for different configurations and ($W=0$) for pooled conditions, is to compensate for the change in the width of the Gaussian kernels?

- Since monkeys are head unrestrained and have natural head movement, the head direction is different in every trial. The authors used head direction info to calculate the projection of the visual stimuli on the retina. Shouldn't one then also consider head-centered encoding of the target as an independent egocentric coding possibility instead of focusing exclusively on target-to-fixation egocentric encoding in their model?

- In the behavioral task "The animal was rewarded for landing its gaze (G) within a circle of radius 8-12° around the original target." Presumably, the reason why animals were rewarded despite such large deviations from the target was to study any biases that the landmark might cause in the saccades. With the described settings, the animal would be rewarded for directing his gaze to any random location between target and shifted location of the landmark. Since the landmark stimulus is quite salient the question is whether the animals actually consider only the 'target stimulus' as the target of the upcoming saccade and whether the landmark was actually behaviorally irrelevant? Could authors rule out the alternative scenario that the monkeys consider both the target stimulus and the landmark as potential saccade targets? More details on the behavior and how the animals were initially trained for this task could be useful. Also, it would be helpful to know what the initial response of the cells to the landmark looks like? Is there spatial tuning to the L positions? And how does this

response change after target presentation?

Specific comments:

p. 3: non-numerical style of in-text citation (Bharmauria et al. 2020; 2021)

p. 4: Vaguely phrased research question: "it remains unclear how landmark-centred visual information is relayed to frontal cortex". Since data from frontal cortex is presented, I do not understand how this research question characterizes the study. It would be nice to formulate a clearer question.

p. 4: model-fitting approach used previously – check if enough info in methods section

p. 9: "the the"

p. 14: "a a"

p. 14 "significantly shifted"

p. 21 broken link "(Fehler! Verweisquelle konnte nicht gefunden werden.b)"

Figures:

Fig. 6 caption typo "described" instead of "describes"

Suppl. Fig. (i): caption refers to a gray box which is missing in the figure

Reviewer #4 (Remarks to the Author):

This study revisited the question of whether visual landmarks influence the visual response of FEF and SEF neurons. The monkeys viewed the onset of a target in the presence of another landmark (in the form of a cross between a vertical and horizontal line). The landmark could appear at different locations relative to the target. Neurons showed visual response fields, and the authors fit the response fields as a function of either target location in retinotopic coordinates or in relation to the relative position of target and landmark. They found that response fields were subtly distorted by the landmarks, resulting in better fits for a hybrid retinotopic-landmark reference frame than a purely retinotopic reference frame.

The paper adds to the prior work and clarifies it.

I would just like to suggest better care in presentation and clarity, as I describe in the specific comments below:

- There are neither page numbers nor line numbers in the manuscript. This makes it very difficult to make specific comments during my review, but I will try my best to point to the correct piece of text that I am commenting on. Please add page numbers and line numbers, at least for first submissions, to help reviewers.

- Introduction, sentence stating "It has been speculated that the visual system codes this influence as target-landmark...": I'm having some difficulty understanding this sentence. Since this concept of "target-landmark" is the main topic of the paper, it would be very nice to clarify this concept already in the introduction. For me, it remains vague what it means at this stage of the paper (i.e. after reading the abstract and first paragraph of the introduction).

- Introduction "However, these studies tested the relative influence on response field activity,": This idea needs to be expanded on and clarified here to highlight the significance of the work.

- Supplementary figures are referred to as (e.g.) Supplementary Fig. 1 but in the later part where they are shown, they are described as Supplementary Fig. i, ii, etc.

- I am having a hard time understanding Fig. 1b because I don't know where the fixation spot is relative to, for example, TLC4.
- The last paragraph of introduction seems to be too technical/detailed and specific (and requiring people to know the details of the prior study in which no landmark effects on visual responses were observed). It also does not end by describing the implications of the new results on the broader question that you started with. The net result is that it makes the paper feel too incremental.
- supplementary fig. 2 legend has several grammatical mistakes.
- Fig. 3b legend seems to have a typo. The upper left (I think) should be TF plotted relative to TF coordinates not relative to TL coordinates.
- Fig. 3c: is the idea of the continuum essentially: that the neural response is fitted as a sum of weight 1 times the TF frame model plus weight 2 times the TL frame model? Saying something like this explicitly is more helpful than panel c1, which is not really very informative (it can be made as a legend in panel c2). It's also a bit odd to label panels a, b, and then c1, c2. Normally, it would be a, b, c, d.
- Fig. 4a: the label "Target onset" occludes data. This is not good. Also, the colorbar is placed in panel a, when it belongs to the other panels.
- the color bar has no quantitative labels. People need to see the kinds of spike counts or spike rates that exist in the data.
- I wonder how useful the black circles are in panel b???? They just self-occlude so much that there is really no information conveyed. For example, I can't really tell how big the circles are, but the size of the circles is exactly what I should be looking for in this figure. The problem is alleviated a tiny bit in the later panels due to the lower numbers of locations, but still...
- the problem is quite worse in supplementary figure 3! The legend of that figure also has some grammatical issues.
- is the raster in Fig. 4a (and corresponding supplementary figure) for all trials or only for the top 10% trials?? Please clarify.
- there are typos in discussion and methods.

Reviewers' comments:

Reviewer #1 (Remarks to the Author):

This paper investigates the spatial frame of reference (FOR) used by the prefrontal cortex to encode visual stimuli locations in the presence of a visual landmark. Specifically, using a modified memory guided saccade task, the authors showed that the visual response of the future saccadic target was encoded in an intermediate FOR between a gaze and a landmark centered FORs. Those results are interesting and in line with findings showing that humans do use visual landmarks to support goal directed behavior. This paper shows a potential neuronal implementation supporting the use of landmarks in the saccadic system.

The authors ability to measure 3d eye and head movements in a head free configuration was critical to be able to distinguish between a gaze and a landmark centered encoding. The authors fitted the visual responses to those 2 FORs and to a range of combination of weighted version of the two. To measure the landmarks effects the authors rightly aggregated together trials with the same Target-Landmark spatial configuration. The results show that the visual responses are better explained by a weighted combination of the gaze and a landmark centered FORs. The use of more landmark centered spatial coding by the frontal cortex was interpreted by the authors as the brain's way to compensate for the variations of the eye and head position in space when the saccadic target was first shown.

The findings are interesting, but the writing style was hard to follow.

RESPONSE: We thank the referee for this accurate summary, positive comments. We have reorganized much of the manuscript and unpacked some of the more obscure sections to make the logic, methods, and results clearer.

Bellow you can find my detailed comments.

RESPONSE: We have attempted to address all of these (See manuscript changes highlighted in Red)

Major comments

1) *The paper needs general improvement in the language used to explain the results: for example, in page 2 in the introduction, line 20, "...nearby may modulate these codes..." the authors do not explain what they mean by modulate. Another example is when explaining the task: page 5, line 3, "...This cross could appear in one..." it seems that the landmark appears after the target is flashed. The authors clearly indicated that the*

landmark is already there when the target is flashed, but the sentence as it is made me confused. Example 3, page 7, line 8, "...variable and arbitrary behavior..." Please explain what would be an example of arbitrary behavior in the current context.

RESPONSE/ACTION: In addition to a general edit throughout the manuscript, we have clarified each of the points raised by the referee:

- 'Modulate these codes' - Clarified on page 2, lines 67-68
- 'This cross could appear' - Clarified on page 6, line 144
- 'Variable and Arbitrary behavior' - Example provided on page 11, lines 263-267

2) My understanding is that once a neuron is isolated, the authors made an initial mapping of the visual receptive field location to estimate its size. Only then, the authors proceeded with the task. The proportion of the visually responsive neurons to the spatially tuned ones is low (for FEF, 102 out of 312; for SEF, only 43 out of 256).

RESPONSE: Yes, we mapped the initial response field of the neuron to locate it and its approximate size. Note that our methods only work on spatially tuned neurons, so we first applied a rigorous statistical test for spatial tuning before sorting our neurons. We recorded a total of 312 FEF and 256 SEF neurons. Of these, 147 (47%, including the visual, visuomotor and motor neurons) FEF neurons (Thompson, 2005; Schall 1991; Bruce and Goldberg 1985) and 79 (31%, including the visual, visuomotor and motor neurons) SEF neurons (Purcell 2010; Schall and Schlag-Rey, 1987) were spatially tuned. These proportions may be slightly lower (because of the rigor of our statistical test) but are generally consistent with the literature. Of those spatially tuned neurons, 101 (69%) FEF and 43 (54%) SEF neurons had visual responses (as opposed to memory/motor only). This is not an unusual proportion (Thompson, 2005; Schall 1991; Bruce and Goldberg 1985; Purcell 2010; Schlag and Schlag-Rey, 1987)

ACTION: This is addressed on page 28, lines 612-619.

The authors suggested that even the non-tuned neurons can potentially contribute to the code. The authors did not speculate how this can be the case in the current study.

RESPONSE/ACTION: We clarified that some decoding studies (which test for implicit codes) have found that they get better results when they include both spatially tuned and untuned neurons in the population analysis (page 32, lines 706-713). However, for our method (which tests explicit spatial coding) we are more confident using neurons with significant spatial tuning.

3) In the discussion, line 8, the authors mentioned that the landmark was behaviorally irrelevant by task design. Indeed, the monkeys were rewarded for making a saccade towards the remembered location and the error window was big enough to include the original location (in 10% of trials) and the shifted location when the landmark shifted (in 90% of trials). Nevertheless, the authors found that the visual response actually shifted towards the landmark, even before the go signal was given. How the authors can rule out

that the monkey's strategy was to encode the target location with respect to the landmark rather than the initial fixation? Ultimately, the landmark was a prominent visual cue on the screen and the target was shown at a fix location (one location out of 4 possibilities) with respect to it. In that case, the landmark might have been used as an anchor and might have been behaviorally relevant as the monkey might have used it to solve the task.

RESPONSE: Apologies, by 'task irrelevant' we meant that the monkeys did not need to use the landmark to get a reward; we did not mean it has no influence on behavior. The behavioral (saccade) data from these two monkeys was already published (Li et al. 2017; Bharmauria et al. 2020, 2021). A stationary landmark was found to reduce variability in final gaze position (Li et. al. 2017). However, when the landmark shifted (after the visual data described here), this caused final gaze distributions to shift almost 1/3 in the same direction relative to the remembered target (Bharmauria et al. 2020, 2021). This suggests that the motor response integrates gaze-centred and landmark-centered coordinates, rather than using one or the other. Likewise, the current study shows that prefrontal visual responses utilize an intermediate eye-to-landmark centered frame. This is consistent with previous studies suggesting that the brain optimally integrates egocentric and allocentric cues (Beck et al., 2008; Kording et al., 2004; Byrne and Crawford 2010).

ACTION: This has been clarified on page 21, lines 469-470. This point about landmark centred coding in the visual system is described in the Discussion (page 21, lines 471-473 and pages 22-23, lines 493-500).

4) The authors did not show how the behavior might be related to the FOR used to encode the saccadic target. A simple way to do so is to correlate the FOR used by the neurons and the FOR used to determine the end point of the saccade.

RESPONSE: This was done in our previous studies (Bharmauria et al. 2020, 2021). As noted above, we could not find any landmark influence on the visual response, but found that later on, (normally eye-centred) motor response field coordinates 1) shift partially in the direction of the landmark and 2) this correlated with the influence on gaze.

ACTION: We have provided this background in the introduction (page 3-5, lines 86-116 and page 5, lines 119-121).

Minor comments

1) The authors did not mention how many cells each monkey contributed with to the data set.

ACTION/RESPONSE: We have provided the details now (page 28, lines 612-619).

2) The authors did not mention how many electrodes were used in each session.

ACTION/RESPONSE: In each session two electrodes were used (one was lowered into FEF and other was lowered in SEF) (page 27, lines 600-602).

3) *Figure 5 top right plot, n=101 must be n=102.*

ACTION: This has been fixed. Actually n equals 101

4) *Page 16 (Discussion), line 13, varable must be variable.*

ACTION: This has been corrected (page 21, line 462).

5) *Page 18, line 19, ‘...the factors...’ must be ‘...these factors....’.*

ACTION: This has been corrected (page 24, line 533).

6) *Page 21, line 8, ‘...then used as to compute...’ must be ‘...then used to compute...’.*

ACTION: This has been corrected (page 27, line 591).

7) *Page 21, line 10, it seems that the authors cited a paper, but the editor did not find the corresponding reference.*

ACTION: This has been corrected (removed).

8) *Page 22, line 6, it seems that the authors cited a paper, but the editor did not find the corresponding reference.*

ACTION: This has been corrected.

9) *Page 24, line 9, ‘.....in eye orientationare....’ must be ‘.....in eye orientation are....’.*

ACTION: This has been corrected (page 30, line 663).

10) *Page 14, line 3, ‘.....to test these results and the single.....’ must be ‘.....to test these results at the single.....’.*

ACTION: This has been corrected. The text has changed substantially at this point.

Nabil Daddaoua

Reviewer #3 (Remarks to the Author):

In their study, the authors investigate the impact of visual landmarks on the neural encoding of visual saccade cues in the frontal lobe areas FEF and SEF. For this, they

carried out an experiment in which rhesus monkeys made saccadic eye movements from a fixation point to the remembered location of a visual target. During the brief presentation of the visual target two intersecting lines were shown on the screen as visual landmark; the visual target briefly flashed in one of the four oblique directions relative to the crossing point of the lines, leading to four target-landmark configurations (TLCs). The cue presentation was followed by a memory period and a mask stimulus. Later the visual landmark reappeared, in some trials randomly shifted in one of the eight radial directions relative to its original position. Monkeys had to saccade to the remembered location of the visual target while their gaze was controlled with unrestrained head movements. A model was fitted to the visual

response of the neurons recorded in FEF and SEF as a function of target location characterized as a 4D vector consisting of its weighted location relative to fixation and relative to the landmark to quantify the relative impact of either frame of reference. The main result is that when the model was fitted to different TLCs separately, a mixed representation of target relative to fixation and relative to landmark resulted in the best fit, i.e. the landmark had a partial impact on cue encoding. Instead, when the model was fitted to pooled data across all target-landmark configurations, target-to-fixation returned the best fit, i.e. landmark had seemingly no impact. The authors concluded that landmark influence depends on the relative positioning of the target and landmark, but not in a geometrically systematic way.

RESPONSE: Thankyou, this is an accurate summary of the original paper.

If I understand correctly, data of this study was published before in REFs 6, 7 (and 27? or just the same paradigm but different datasets were used before – please clarify). Previous studies had a focus on other time periods in the trial. The current study revisits the question of landmark-relative visual encoding, asking if a TLC-specific analysis of visual cue responses yields additional insights compared to the TLC-pooled analyses; in the latter, a gaze-centered (“target-to-fixation”) encoding was observed before.

RESPONSE: These were the same experiments, and the dataset is largely the same, but in our previous studies (Bharmauria et al. 2020, 2021) we focused on influence of a landmark shift on the later memory and motor activity of FEF and SEF respectively. We did not find any influence of the stationary landmark on the visual response. It now appears that this was because 1) we did not have a measure sensitive enough to detect a partial influence of a stationary landmark, and 2) we pooled the data across stimulus configurations. Here, we used a more sensitive continuum between eye-centered and landmark-centered coding and applied this separately to different target-landmark configurations, to successfully detect a configuration-dependent, intermediate code.

ACTION: As requested by the editor, we have clarified the relationship between our current study and the previous studies in the introduction (pages 3-5, lines 86-116) and discussion (pages 20-21, lines 453-458). We have stated explicitly that this is largely the same dataset, but the hypotheses and analysis are new (page 5, lines 117-130).

(See manuscript changes highlighted in Red)

The research is potentially interesting to researchers of visual sensory processing and higher cognitive functions such as spatial cognition and cognitive motor functions. The research topic of allocentric encoding has never lost its timeliness, since fundamental aspects of this important form of spatial cognition remain unanswered until today. The current study has the potential to add detail to previous findings,

RESPONSE: Thank you very much for these positive comments.

but at this stage, the somewhat curious observations presented seem not to provide the major step forward in our deeper understanding.

RESPONSE: To our knowledge, this is the first time that the influence of an independent visual landmark on prefrontal visual responses has been successfully demonstrated, at least in a sensorimotor task. The closest work that we know of is Carl Olson lab's work on object-centered coding in SEF, but they did not study the visual response and did not look at the influence of an independent landmark. We hope that after our revisions the referee will agree that this is a novel contribution to systems/computational neuroscience.

ACTIONS: We have expanded considerably on background and functional significance of these data in the Introduction (pages 3-5, lines 86-116) and discussion (page 20-21, lines 453-458 and pages 22-23, lines 488-500), moved one supplementary figure to main results (now Figure 2) to show how landmarks and eye torsion interact in this specific paradigm, provided the additional analysis requested, and have provided a more rigorous and complete statistical analysis of the individual neuron and population data relative to controls (Figures 6 and 7). See details below.

Main comments:

- To say it in the words of the authors: "It is important to note that this TF-TL shift [i.e. the evidence for partial landmark-relative encoding] does not mean response fields shifted toward the landmark, which might indicate a simple shift in the perceived target direction. The small response field shifts that we observed (e.g., Figure 4 c3) were highly variable and did not follow any clear pattern across neurons." Having said this, the authors should provide convincing evidence that their model is not just tracking noise. They try so by comparing against shuffled surrogate data, but it is not fully clear to me, how with the N=100 randomizations they exactly to the statistically testing for significance of the observed result against the surrogate. More detail will be helpful to convince.

RESPONSE: In this analysis we shuffled the relationship between landmark location and neural activity for each cell 100 times. It is reasonable to assume that noise present in the real data would remain in the surrogate randomized data, whereas the signal should be gone. If our results were caused by noise, they should be produced by both datasets. Conversely, if our results were caused by real signals, they should disappear in the surrogate data (which is what we observed). We went for a sample size of N=100 since

it is the smallest magnitude of sample size that assures a stabilized median (a desirable outcome), but higher numbers of iterations yielded the same result.

ACTIONS:

As requested, we have provided more detail for this analysis, both in the methods and results sections. This has become the centrepiece of the results section, specifically:

- We have added this logic to the results text (page 14, lines 331-336)
- We have expanded on the description in the methods (page 33-34, lines 733-742).
- We have replaced the ‘violin plot’ summary of the pooled data with a more complete analysis of real vs. shuffled data at both the population and individual unit level (new Figure 6). We show that at the population level, there is no significant difference in the means or medians of the TF-TL coordinate continuum between real and shuffled data in either FEF (Fig. 6a, b). In addition, we show that there is a significant difference in 80.2% of FEF and 67.4% SEF individual neurons (Fig. 6c, d), but these differences were bidirectional (toward or away from landmark-centered coding) and thus apparently cancel out at the population level.
- Note that in comparison 100% of FEF neurons and 90.7% SEF neurons showed a significant landmark influence for at least some individual target-landmark configurations, and the FEF distribution showed a significant shift toward landmark-centered coding (Figure 7).
- We have similarly expanded the analysis of the separate target-landmark configuration analysis at the population level (Fig. 7a, b) and single unit level (Fig. 7 c, d).

At this stage the conclusions seem unsatisfying, since they suggest that in a previous publication a non-justified pooling was applied, washing out potentially interesting effects; but now that a more appropriate approach is taken, it is still not really clear what the impact of the landmark is all about.

RESPONSE: The functional significance of such landmarks is that they provide additional information that improve performance in the presence of noisy egocentric signals (Li et al. 201; Byrne et al 2010). It has previously been shown that the presence of stable visual landmarks improves pointing accuracy in a memory guided pointing and gaze tasks (Byrne et al. 2010). The latter study employed the same animals and task reported here. The question addressed in the current study is if and how this information is relayed to the visual response in prefrontal motor areas – a question that has never been answered before.

ACTIONS:

- We have expanded the description of the functional significance of landmarks for gaze in the Introduction (pages 3-4, lines 89-95).

- Note that there is also considerable discussion of this issue on pages 22-23, lines 488-500.

I wonder if this the right stage of the study to publish it.

RESPONSE: We hope the referee changes their mind after these revisions.

-- Side note: The summarizing statement at the beginning of discussion "the landmarks caused a shift in the reference frame for coding the target toward the landmark (TL)" seems to contradict the above statement that results do NOT mean a RF shift toward the landmark. Please phrase such that confusion is avoided.

RESPONSE: We said this to highlight that we are reporting reference frame shifts. There is relationship between the two: if the reference frame was completely fixed and the only factor influencing response fields (and all other conditions were held constant) one might expect a similar shift in the response field. However, our algorithm only provides the best overall frame of reference for the whole response field. It does not detect details, e.g., interactions between the landmark and different stimulus locations. When one factors in various other factors that might influence the response field (e.g., distributions of initial eye position, landmark position, specific locations on the RF, trial history, trial-to-trial variations in attention to target vs. landmark) its not surprising that they do not follow such a simple rule.

ACTIONS:

- We have clarified this language in the discussion (page 21, lines 459-468)
- We have noted in the introduction that some studies have looked at RF shifts in frontal visual RFs due to surrounding distracters but did not look at their influence on the underlying coordinate frame (page 3, lines 83-85).
- We have explained the methodological points described above more fully in the methods (page 30, lines 658-662), section: fitting neuronal response fields and spatial models) and alluded to this in the discussion (page 21, lines 459-462).

- As explained in the methods, "The bandwidth of the Gaussian kernel (KW) was determined for each neuron individually to match the response field's size, shape, and contour. This was done by calculating the PRESS statistic for each spatial model for all bandwidths between 1 and 15. Then the bandwidth yielding the lowest residuals was deemed as the best fit or spatial model." It is not clear whether the best models have different (KW) for different target-landmark configurations and pooled conditions? If that is the case, how did authors rule out the possibility that the reason why (0

RESPONSE: There appears to be something missing or cut off at the end of the referee's comment, so we are not clear what the question was. However, we have provided details about the choice of KW in the methods (page 29, lines 648-652).

- Since monkeys are head unrestrained and have natural head movement, the head direction is different in every trial. The authors used head direction info to calculate the projection of the visual stimuli on the retina. Shouldn't one then also consider head-centered encoding of the target as an independent egocentric coding possibility instead of focusing exclusively on target-to-fixation egocentric encoding in their model?

RESPONSE: Actually, we used direct recordings of eye orientation in space to compute target location relative to the eye. Nevertheless, we already tested between eye, head, and body/space centered models in this dataset (Bharmauria et al. 2020, 2021). At the overall population level, the eye-centered model gave significantly better fits than the head and body-centered models in both visual and motor responses. In order to make a clear comparison with the landmark-centred model at the population level we chose the overall best model (target-in-eye, or TF as we call it here). We acknowledge there may be more complexity at the single cell level, which we hope to investigate in another study.

ACTIONS:

- We have clarified how target-in-eye positions were calculated (page 28, lines 627-628, equation i)
- Rather than duplicate the statistical analysis from the previous paper, we have summarized it with a new supplementary figure iii (page 46) comparing the PRESS residuals between target in eye (Te), target in head (Th) and target in space (Ts) coding for FEF and SEF. This analysis shows that the PRESS residual for Te is significantly lower than both Th and Ts coding. This analysis has also been summarized in the text (page 10, lines 247-249) with justification for using Te/TF as the best representative model for comparison with TL.

In the behavioral task "The animal was rewarded for landing its gaze (G) within a circle of radius 8-12° around the original target." Presumably, the reason why animals were rewarded despite such large deviations from the target was to study any biases that the landmark might cause in the saccades. With the described settings, the animal would be rewarded for directing his gaze to any random location between target and shifted location of the landmark. Since the landmark stimulus is quite salient the question is whether the animals actually consider only the 'target stimulus' as the target of the upcoming saccade and whether the landmark was actually behaviorally irrelevant? Could authors rule out the alternative scenario that the monkeys consider both the target stimulus and the landmark as potential saccade targets? More details on the behavior and how the animals were initially trained for this task could be useful.

RESPONSE: We used these large reward windows (surrounding the target, with the landmark intersection point within one edge) to avoid influencing behaviour toward or away from the landmark. With the head-unrestrained range of motion and large screen (2.2 m X 1.6 m) they still had to look toward the target to get the reward.

As we reported previously (Li et al. 2017), there was a small (15% of the landmark shift) systematic bias of final gaze position in direction of the landmark shift, but overall gaze correlated much better with target position than with landmark position.

ACTIONS: We repeated this analysis and report it on page 6, lines 149-153. A systematic bias of final gaze toward landmark was noticed (27% toward the landmark in Monkey L / 15% Monkey V) and correlations to landmark positions were lower (0.11 Monkey L, 0.06 Monkey V) compared with target correlations (0.72 Monkey L, 0.68 Monkey V).

We have also noted in the discussion (page 22, lines 482-487 and 493-495) that landmark biases in the visual response may have contributed to these behavioral biases, although ultimately, they are most directly related to the motor responses analyzed in our previous studies (Bharmuria et al. 2020, 2021).

Also, it would be helpful to know what the initial response of the cells to the landmark looks like? Is there spatial tuning to the L positions? And how does this response change after target presentation?

RESPONSE: See the following mean spike density curves for our FEF and SEF populations. There was only a very low and ‘sluggish’ response to landmark onset (-1000), perhaps more preparatory than visual. This dissipated somewhat before target onset (time 0). Approximately 80 ms later there was a robust, significant target-related response. Only the data in the blue analysis window is described in the current paper.

ACTION: We have added this figure in the supplementary section (fig ii) section, and it has been reported in text (page 9, lines 209-211).

Specific comments:

p. 3: non-numerical style of in-text citation (Bharmauria et al. 2020; 2021)

ACTION: The references have been fixed.

p. 4: Vaguely phrased research question: “it remains unclear how landmark-centred visual information is relayed to frontal cortex”. Since data from frontal cortex is presented, I do not understand how this research question characterizes the study. It would be nice to formulate a clearer question.

RESPONSE: We were posing the question: if landmarks do not influence visual inputs to prefrontal cortex (which we failed to detect previously) how could they subsequently influence prefrontal memory / motor responses (which we did find previously)? In other words, where is this landmark information coming from if not in visual inputs? This was our motivation for looking more closely at the FEF / SEF visual response.

ACTION: We have clarified the question(s) (page 4-5, lines 109-116).

p. 4: model-fitting approach used previously – check if enough info in methods section

ACTION: We have provided substantial details to the approach.

p. 9: “the the”

ACTION: This has been removed.

p. 14: “a a”

ACTION: This has been removed.

p. 14 “significantly shifted”

ACTION: This has been corrected. The text has substantially changed at this point in the text.

p. 21 broken link “(Fehler! Verweisquelle konnte nicht gefunden werden.b)”

ACTION: This has been removed.

Figures:

Fig. 6 caption typo “described” instead of “describes”

ACTION: This has been corrected. Again, the text has substantially changed.

Suppl. Fig. (i): caption refers to a gray box which is missing in the figure

ACTION: This has been removed from the caption.

Reviewer #4 (Remarks to the Author):

This study revisited the question of whether visual landmarks influence the visual response of FEF and SEF neurons. The monkeys viewed the onset of a target in the presence of another landmark (in the form of a cross between a vertical and horizontal line). The landmark could appear at different locations relative to the target. Neurons showed visual response fields, and the authors fit the response fields as a function of either target location in retinotopic coordinates or in relation to the relative position of target and landmark. They found that response fields were subtly distorted by the landmarks, resulting in better fits for a hybrid retinotopic-landmark reference frame than a purely retinotopic reference frame.

RESPONSE: This is generally correct, although technically we tested the continuum of weights between eye-centered and landmark-centered coding and found a shift along this continuum.

The paper adds to the prior work and clarifies it.

RESPONSE: Thank you for the positive assessment and feedback. We have addressed each of the specific comments as described below:

I would just like to suggest better care in presentation and clarity, as I describe in the specific comments below:

RESPONSE: Thank you for the editorial comments, we have reviewed the paper and made the specific corrections detailed below.

- There are neither page numbers nor line numbers in the manuscript. This makes it very difficult to make specific comments during my review, but I will try my best to point to the correct piece of text that I am commenting on. Please add page numbers and line numbers, at least for first submissions, to help reviewers.

ACTION: We have added the page and line numbers in the revised version.

Introduction, sentence stating “It has been speculated that the visual system codes this influence as target-landmark...”: I’m having some difficulty understanding this sentence. Since this concept of “target-landmark” is the main topic of the paper, it would be very nice to clarify this concept already in the introduction. For me, it remains vague what it means at this stage of the paper (i.e. after reading the abstract and first paragraph of the introduction).

ACTION: Here we are referring specifically to the spatial relationships between landmark and the target. We have defined it in the manuscript now (page 3, lines 74-75).

- *Introduction “However, these studies tested the relative influence on response field activity,”: This idea needs to be expanded on and clarified here to highlight the significance of the work.*

ACTION/RESPONSE: Some studies have investigated response field shifts in frontal visual responses due to surrounding distracters but did not look at their influence on the underlying coordinate frame. We have further clarified it in the manuscript now (page 3, lines 83-85).

- *Supplementary figures are referred to as (e.g.) Supplementary Fig. 1 but in the later part where they are shown, they are described as Supplementary Fig. i, ii, etc.*

ACTION: We have corrected this throughout the text.

I am having a hard time understanding Fig. 1b because I don't know where the fixation spot is relative to, for example, TLC4.

RESPONSE/ACTION: This (Fig. 1b) is a ‘blow-up’ around the region of the target (Fig 1.a, as shown in orange rectangle), and thus does not include the fixation point. We have added the orange rectangles (Fig 1a and b), and this has also been clarified in the figure legend.

- *The last paragraph of introduction seems to be too technical/detailed and specific (and requiring people to know the details of the prior study in which no landmark effects on visual responses were observed). It also does not end by describing the implications of the new results on the broader question that you started with. The net result is that it makes the paper feel too incremental.*

ACTION: We have removed the technical jargon (TF, TL etc) and highlighted the hypotheses (pages 5, lines 119-124) more clearly in this paragraph.

- *supplementary fig. 2 legend has several grammatical mistakes.*

ACTION: We have fixed this (and it is now moved to the main paper).

- *Fig. 3b legend seems to have a typo. The upper left (I think) should be TF plotted relative to TF coordinates not relative to TL coordinates.*

ACTION: We have fixed this.

- *Fig. 3c: is the idea of the continuum essentially: that the neural response is fitted as a sum of weight 1 times the TF frame model plus weight 2 times the TL frame model? Saying something like this explicitly is more helpful than panel c1, which is not really very*

informative (it can be made as a legend in panel c2). It's also a bit odd to label panels a, b, and then c1, c2. Normally, it would be a, b, c, d.

RESPONSE/ACTION: We have added panels a,b,c now.

- Fig. 4a: the label "Target onset" occludes data. This is not good. Also, the colorbar is placed in panel a, when it belongs to the other panels.

ACTION: We have put the label on top of the Figure now and the color bar is shown to the right now.

- the color bar has no quantitative labels. People need to see the kinds of spike counts or spike rates that exist in the data

ACTION: We have labelled the colour bar now.

- I wonder how useful the black circles are in panel b???? They just self-occlude so much that there is really no information conveyed. For example, I can't really tell how big the circles are, but the size of the circles is exactly what I should be looking for in this figure. The problem is alleviated a tiny bit in the later panels due to the lower numbers of locations, but still....

RESPONSE: These circles are important because they represent the actual data (circles show the magnitude of the neural response superimposed on its position in the coordinate frame of the plot). It is the difference between these data and the nonparametric fits (shown as the background color plot) that we use to calculate goodness of fit. This is described in the text (page 12, lines 299-305).

ACTION: We agree that the 'circles' were too dense in the original figure (now Figure 5), because too much data was superimposed. To alleviate this, we have split this figure into two: Fig 5b and 5c. 5b shows the magnitude of responses as circles and 5c shows the corresponding non-parametric fit to these circles.

- the problem is quite worse in supplementary figure 3! The legend of that figure also has some grammatical issues.

RESPONSE/ACTION: We have redone the supplementary figure (now figure iv) along the lines of figure 5 in the main text. We have also corrected the legend.

- is the raster in Fig. 4a (and corresponding supplementary figure) for all trials or only for the top 10% trials?? Please clarify.

RESPONSE: The raster in 4a (now 5 a) represents top 10 % trials. However, spike density plots are done for both top 10 % trials (red) and all the trials (dark red).

ACTION: This has been clarified in the figure legend.

- there are typos in discussion and methods.

ACTION: These have been fixed using a spell/grammar check.

Reviewers' comments:

Reviewer #1 (Remarks to the Author):

The authors did address the concerns that I raised during the first revision round. The language of the paper is improved, more references were added to support the authors' claims. The authors did clarify the concern that I have about the behavioral task and the possible monkey strategy to solve it. All together, I support the current version of the paper for publication.

Nabil Daddaoua

Reviewer #3 (Remarks to the Author):

This is a substantially revised version of previously submitted manuscript. I appreciate the major effort that the authors invested, especially in explaining more background. The extensive added explanations clearly aimed at addressing previous concerns and help to make the novelty more obvious and make the rationale of the approach more accessible. Unfortunately, the new information does not fully resolve all major methodological concerns.

I still have open questions regarding the data quantification. The main conclusions of the paper hinge on a comparison between empirical sample and shuffled data as control, for example when testing for shifts of the frame of reference in individual neurons. In lines #350ff, what exactly is compared with what for each neuron with the ranksum test? The explanations in Methods (#729ff) do not fully resolve this question. If all analyses are repeated for the 100 shuffles, then for each result it should be directly decided if the empirically observed value is larger/smaller than 95% of the outcomes from the shuffled data (by simply counting the cases). For such test, 100 permutations would barely be enough. Yet, it seems that this is not what is done, since a ranksum test compares two distributions. But here only one distribution exists (the 100 shuffles) and a single reference value (the observed empirical value). Or do the authors mean a signed rank test which compares the median of one distribution (shuffle) against the fixed value (observation)? Since the power of the signed rank test also depends on the sample size and the sample size of the shuffle distribution is arbitrarily chosen, I think this is not the correct approach for a permutation test. I am questioning this because it is not plausible to me how neurons can differ significantly from the shuffled data when they fall almost perfectly on the unity line in Fig. 6/7? Is there a parameter ranked other than the visualized coordinates in the plot (TF-TL continuum)? A full understanding of the procedures and their adequacy will be necessary to decide if the results are meaningful.

Similarly, I find the vertical position of the median in Fig. 7a highly implausible given the depicted data: I am counting 11(13) data points above, and multiple times more below. How can this be? The result could only be true if most of the data in the upper range was invisible (e.g. hidden behind the text box or out of range - both would be an absolute "no-no"). Further, I assume that the shift of the median compared to the shuffled data is quantified as above, i.e., here by comparing the median of the medians in the shuffled data against the observed median? Please state explicitly as #408 does mention the type of test that is performed.

I apologize that part of a previous question apparently got lost somewhere on the way. Here is the full original question: It is not clear whether the best models have different (KW) [kernel width] for different target-landmark configurations and pooled conditions? If that is the case, how did authors rule out the possibility that the reason why ($0 < W < 1$) returns the best fit for different target-landmark configurations as compared to ($W=0$) for pooled conditions, is to compensate for the change in the

width of the Gaussian kernels?

The authors report "SEF responses were generally more broadly tuned". Could this broader tuning compared to FEF neurons, explain why on average SEF did not show a significant shift towards TL in Fig 7b? Is it possible that this landmark stimulus was not optimal for driving SEF neurons? Or, perhaps the distances between the target and landmark, relative to the width of the receptive fields were not comparable to FEF neurons? This question might be answered by carefully analyzing the responsive field of those SEF neurons in Fig 7b which deviated from the unity line and comparing them with response fields of the FEF neurons.

Currently, the result presentation is rather abstract and the examples neurons seem quite detached from the remaining analyses done with the fitting procedures. This makes assessment of the results additionally difficult. Since the main conclusion is about landmark effects on individual neurons, it should be possible to directly use example neurons for illustrating the main finding and the subsequent steps of analysis. The example in Fig. 3 does not help me in this regard. Maybe presenting intermediate results of the model fitting (e.g., w parameter) for the example neuron (and its shuffle data in the supplement) would help? Minimum, indicating which data point in Fig. 6/7 corresponds to the example neurons from FEF and SEF would help get a better impression of the data. What makes this part also difficult is that the new section "model fitting approach" in Results is hard to follow. There is mentioning of "our model-fitting analysis", "egocentric analysis from previous study", "new method" and "standard model-fitting method" and it is not clear what exactly was done and which of these procedures are identical, or which phrases refer to different procedures. It might be just a matter of rephrasing, but I am not sure if enough information is contained in the paper to answer this. In this context, I am also wondering about the schematic in Fig. 4. The way I understand it, it shows simulated or even fictitious data/fits (although, in #700 it says "example"?). If the idea of this figure is to convince the reader of the new model fitting approach, wouldn't it be more appropriate to demonstrate the fitting procedure with real example neurons, e.g. the one from Fig. 5, and its shuffled control to demonstrate its sensitivity?

Interpretation of the results: In response to previous reviewer comments, the authors expanded on the argument that the results do not (necessarily) indicate a shift of the RF towards the landmark, i.e., not a spatial shift (#459). I still find that then statements like the one in #468 ("shifts toward TL") are potentially confusing - I guess what is meant with the "shift" here is not a spatial shift but that the TL model compared to the TF model gains more weight? If this meant then I find it still confusing that in the next sentence it is argued that the observed shift might explain gaze shifts - which would be indeed spatial shifts?

In general, I find the writing still worth improving; often terms and concepts are not well introduced and references to previous paragraphs/concepts/thoughts remain unspecific (see examples below), making it overall a hard read. The paragraphs in the introduction about own previous work (added in response to previous reviewer requests) partly read like a narration (we did this, then we did that...) rather than a succinct line of argumentation and putting into context of the research question. Reviewer 1 had already complained in the previous review about writing style; especially the new paragraphs would benefit from thorough internal revisions; some specific examples of what I mean can be found below.

Specific points:

#67 "stabilizing noise" - does not make sense to me; did you mean "stabilizing against noise fluctuations" or something similar?

#96 "this process" - reference unclear (which process is meant?)

#97 "this task" - again, reference unclear; please be specific

#100 "this landmark influence first appeared as a multiplexed signal in the memory response, and then became fully integrated into the final motor code" - sound a bit like lab-slang - "multiplexed" in what sense? what is meant by "motor code" (the explanation in parentheses does not help)?

#123 "a relatively unprocessed [...] manner" - meaning unclear

#124 "to do this" - do what (previous sentence says "we hypothesized")?

#134 "for more precise behavior" - is there a performance measure for motor accuracy that the authors are referring to? Was this introduced already?

#150 "final gaze" - undefined; which time period is analyzed here?

#150 "27%" - of what, the target-landmark distance ("15%" correspondingly)? Please specify.

#255 "we also tested other" - please specify which other continua you tested

#306 "variability [...] persists due to non-spatial factors such as attention and motivation" - speculative, rather a point of discussion than results

#324 the main text does not make reference to Fig. 5d-g (and the corresponding figure caption is generic). What am I supposed to learn from the results shown in these panels?

#353 "(Fig. 6c" - parenthesis missing

#705/712 "tests the explicit spatial codes used by neurons" - no, we do not know what neurons are actually using; it is just a fitting procedure.

#715 please introduce PRESS

#736 "stabilized median" - unclear, median of what? Also, if the computations were done with higher number of shuffle iterations, as the sentence suggests, then why not show the result with the higher numbers, which would be statistically safer?

Caption Fig. 3: "more robust plots" - I guess you mean responses, not plots

Caption Fig. 6: It is not explained in which sense this is a swarmchart, since it looks like a regular scatter plots. More important, if both the x- and the y-axis are meaningful and the distribution width and deviations from unity line are main results then I do think that jittering the data for better visibility is justified.

Fig. 3 and 5/data sampling: Since the grand average curves in Fig. 3b and the corresponding example data (e.g. Fig. 5a) contain the top 10% of responses, one wonders if any response would be left, when looking at only the remaining 90% - probably not, but one cannot tell since the second curve contains also the top 10%; if the 90% "residuals" are flat, does it still make sense to include this (likely very noisy) data in the analyses?

Fig. 6/7: Please do not overlap data with text boxes.

Fig. 6: Why TL(C1) label in c/d, isn't this pooled data?

Reviewer #4 (Remarks to the Author):

This version reads significantly better than the previous version.

Some additional comments:

- Supp. Fig. iv: panel a: I still do not understand why the authors like to obscure data points with the color legend. The spikes are hidden by the color legend. Also, please explain in the figure caption whether the shown raster is for the top 10% of trials or whether it is for all trials.

- panels b-e: I'm also not sure what the label of the colorbar means. What does Normalised neural activity mean? By normalisation, did you baseline-subtract or what exactly? i.e. in what sense is it "normalised"? Please clarify

- Fig. 5: similar comments to the above (except the top 10% raster, which is now clearly described for this figure)

- Fig. 6a, b: again, the legends look like they may hide data points. Also, what does "signi units" and "not signi units" mean???

- there are still too many typos, missing parentheses etc for my taste

Overview of major manuscript changes:

Changes made in direct response to referees' comments are highlighted in yellow. These changes are detailed in the following sections.

Changes that resulted from the editor's comments are highlighted in blue. A persistent comment was that we need to differentiate this new analysis from our previous work. The editor asked us to repeat the previous analysis in the new paper. We did this and confirmed that, overall, the population preferentially encodes the target relative to the eye (Figure 5). However, we felt that (to be consistent with the rest of the paper and the referees' comments) we should analyze this data with the same scrutiny that we did elsewhere in the paper, including individual neuron analysis and intermediate coding analysis. This resulted in several new findings that were both robust and important:

- When we applied the same test to individual neurons (new Fig. 6), we found that nearly as many FEF neurons code the landmark (30%) vs. the saccade target (40%).
- To test this for intermediate codes (like we did for reference frame coding), we developed a new spatial continuum between target and landmark coding (Figure 7, top row). This showed that many neurons showed intermediate target-landmark codes, especially when we analyzed separate landmark configurations.
- This further allowed us to test if target-landmark interactions could also account for the shift toward landmark-centered target coding (Fig. 7, bottom row). The answer was yes: neurons that coded both targets and landmarks also showed the biggest shift toward landmark-centered target coding, in both SEF and FEF (Fig. 8).

We think this is a much stronger paper, because it shows prefrontal coding of both target and landmark information and provides a novel cellular mechanism for landmark-centered target coding. We suggest this may be the general mechanism for ego/allocentric visual integration for action.

Reviewers' comments:

Reviewer #1 (Remarks to the Author):

The authors did address the concerns that I raised during the first revision round. The language of the paper is improved, more references were added to support the authors' claims. The authors did clarify the concern that I have about the behavioral task and the possible monkey strategy to solve it.

All together, I support the current version of the paper for publication.

Nabil Daddaoua

REPLY: We thank the reviewer for their feedback throughout the review process and supporting it for publication.

Reviewer #3 (Remarks to the Author):

This is a substantially revised version of previously submitted manuscript. I appreciate the major effort that the authors invested, especially in explaining more background. The extensive added explanations clearly aimed at addressing previous concerns and help to make the novelty more obvious and make the rational of the approach more accessible. Unfortunately, the new information does not fully resolve all major methodological concerns.

REPLY: We thank the reviewer for their comments on the manuscript and they have further helped in the improvement of the manuscript. We have addressed the concerns of the reviewer and as described above highlighted these in yellow in the text. We have also corrected the figures as suggested.

One difficulty with the previous paper was that the results were largely statistical and not clear to see in the figures. The new version addresses this in two ways. First, the new results reported (Fig. 6, Fig. 7 top row, Fig. 8) are generally more robust than the original results (see above). Second, we have plotted more data. Specifically, in the previous 'continuum plots' we went from plotting the original configuration-pooled data fits directly to 'recombined' fits. Now, we have combined this data in one plot (Figure 7) and plotted the intermediate configuration-dependent data in the middle column. Here it is easier to see that the configuration-dependent fits show a broad distribution of fits relative to controls, so it is perhaps clearer how there is a significant increase in distribution relative to controls. One can also see that after recombining these data through averaging fits, there is a systematic population shift toward landmark coding (Fig. 7e). Other more detailed statistical effects are described in the text.

I still have open questions regarding the data quantification. The main conclusions of the paper hinge on a comparison between empirical sample and shuffled data as control, for example when testing for shifts of the frame of reference in individual neurons. In lines #350ff, what exactly is compared with what for each neuron with the ranksum test? The explanations in Methods (#729ff) do not fully resolve this question. If all analyses are repeated for the 100 shuffles, then for each result it should be directly decided if the empirically observed value is larger/smaller than 95% of the outcomes from the shuffled data (by simply counting the cases). For such test, 100 permutations would barely be enough. Yet, it seems that this is not what is done, since a ranksum test compares two distributions. But here only one distribution exists (the 100 shuffles) and a single reference value (the observed empirical value). Or do the authors mean a signed rank test which compares the median of one distribution (shuffle) against the fixed value (observation)? Since the power of the signed rank test also depends on the

sample size and the sample size of the shuffle distribution is arbitrarily chosen, I think this is not the correct approach for a permutation test.

REPLY/ ACTIONS:

We would like to express our gratitude to the reviewer for bringing up these important points. In response, we have conducted a revised statistical analysis. Instead of using the previous analysis method (sign rank test), we now employed bootstrapping to assess the difference between the empirically observed value and the median of shuffled data (100 shuffles) on a single neuron basis. The details of this new approach are explained in the methods section (Pages 42-43, lines 956-973).

It is important to note that the statistical power of this analysis remains unaffected by the number of shuffles performed. This holds true for both the original data (depicted in Figure 7 lower row) and the new continuum analysis data presented in Figure 7 (upper row). For the population data, we compared the median of the medians derived from the shuffled data with the median of the observed data using the Wilcoxon signed rank test.

For the configuration-dependent analysis, the same procedure was followed.

I am questioning this because it is not plausible to me how neurons can differ significantly from the shuffled data when they fall almost perfectly on the unity line in Fig. 6/7? Is there a parameter ranked other than the visualized coordinates in the plot (TF-TL continuum)? A full understanding of the procedures and their adequacy will be necessary to decide if the results are meaningful.

REPLY: We looked at units that display a significant difference despite being close to the unity line in more detail. These units are significant because they show very little variance, i.e., zero scattering between shuffles and bootstraps. This is most likely due to the discrete nature of the continuum analysis only allowing for multiples of 0.1.

This is mentioned on Pages 42-43, lines 965-973.

Similarly, I find the vertical position of the median in Fig. 7a highly implausible given the depicted data: I am counting 11(13) data points above, and multiple times more below. How can this be? The result could only be true if most of the data in the upper range was invisible (e.g. hidden behind the text box or out of rang - both would be an absolute "no-no").

REPLY: We thank the reviewer for pinpointing this. The distributions have been corrected (now Figure 7 h). We have also carefully checked all other figures in the main text as well as in the supplementary section.

Further, I assume that the shift of the median compared to the shuffled data is quantified as above, i.e., here by comparing the median of the medians in the shuffled data against the observed median? Please state explicitly as #408 does mention the type of test that is performed.

REPLY: Yes, the reviewer is right. We have compared the median of the original data against the median of the medians of the shuffled data, using the Wilcoxon signed rank test. This has been stated explicitly on Page 21, lines 473-478.

I apologize that part of a previous question apparently got lost somewhere on the way. Here is the full original question: It is not clear whether the best models have different (KW) [kernel width] for different target-landmark configurations and pooled conditions? If that is the case, how did authors rule out the possibility that the reason why ($0 < W < 1$) returns the best fit for different target-landmark configurations as compared to ($W = 0$) for pooled conditions, is to compensate for the change in the width of the Gaussian kernels?

REPLY/ACTION: We thank the reviewer for bringing this point up. The optimal kernel width was determined once for each neuron. The same kernel width was used for each reference frame and each target-landmark configuration as well as the pooled condition.

This has been clarified in the section 'Fitting neuronal response fields against spatial models' and has been clearly stated on Page 38, lines 875-877.

The authors report "SEF responses were generally more broadly tuned". Could this broader tuning compared to FEF neurons, explain why on average SEF did not show a significant shift towards TL in Fig 7b? Is it possible that this landmark stimulus was not optimal for driving SEF neurons? Or, perhaps the distances between the target and landmark, relative to the width of the receptive fields were not comparable to FEF neurons? This question might be answered by carefully analyzing the responsive field of those SEF neurons in Fig 7b which deviated from the unity line and comparing them with response fields of the FEF neurons.

REPLY/ACTION: We reported this general trend by simply visualizing neurons, however, because of the fragmented/scattered nature of the hot-spots of response fields, we could not derive a simple measure to quantify this. In the paper (Pages 16-17, lines 392-398) we report it as: SEF neurons were generally more broadly tuned, however, due to the fragmented, scattered nature of 'hot spots' in some FEF neurons and many SEF neurons, it was difficult to derive a simple measure (such as bandwidth) to compare their response fields

Currently, the result presentation is rather abstract and the examples neurons seem quite detached from the remaining analyses done with the fitting procedures. This makes assessment of the results additionally difficult. Since the main conclusion is

about landmark effects on individual neurons, it should be possible to directly use example neurons for illustrating the main finding and the subsequent steps of analysis. The example in Fig. 3 does not help me in this regard.

ACTION: First, we have expanded Figure 2 and the accompanying text (Page 7, lines 170-180) to provide more rigorous definitions of the models we are talking about and repeat these models in Figure 4a. Second, we have simplified figure 4 and gone through the steps of calculating residuals (Pages 10-13). First, we illustrate the simulated data (4b) to present the logic of our methodology in the clearest way (differences in actual residuals are often hard to see with the naked eye and require statistical analysis). Fig. 4d then shows the non-parametric fit that yielded the lowest residuals, in this case Target relative to Fixation in eye coordinates [$T_{F(e)}$]. Then Fig 4e shows the actual data superimposed on the fit, with residuals between them shown on the side. (The continuum analysis has now been moved to Figure 7 and supplementary figure v.)

Maybe presenting intermediate results of the model fitting (e.g., w parameter) for the example neuron (and its shuffle data in the supplement) would help?

ACTION: At this point in the paper, we are contrasting fits between different models, rather than the shuffled data (the shuffle controls only come later in Figure 7, so we will return to this below). Instead, to address this comment we have contrasted a target favoring response field in eye coordinates (Fig. 6a) to another example response field that showed a best fit to landmark-relative to fixation in eye coordinates (Fig. 6c). This figure also provides the statistics for these two neurons and summarizes the best fits for all neurons (Fig. 6b, d).

Minimum, indicating which data point in Fig. 6/7 corresponds to the example neurons from FEF and SEF would help get a better impression of the data.

ACTION: As requested, we have highlighted our two example neurons in Figure 7 c/f so that readers can cross-reference the data.

What makes this part also difficult is that the new section "model fitting approach" in Results is hard to follow. There is mentioning of "our model-fitting analysis", "egocentric analysis from previous study", "new method" and "standard model-fitting method" and it is not clear what exactly was done and which of these procedures are identical, or which phrases refer to different procedures. It might be just a matter of rephrasing, but I am not sure if enough information is contained in the paper to answer this.

REPLY/ACTION: This should be clear now. As requested by the editor, we have repeated our original analysis (Bharmuria et al. 2020, 2021), in which we pooled all of

the neurons and tested between models, in Figure 5 and Pages 14-15, lines 298-346. Once again, this seems to suggest the responses just code target relative to eye. Then do a new neuron-by-neuron analysis (Fig. 6) and finally explained the new continuum analysis (Pages 17-18, lines 411-423). This is followed by analysis of two spatial continua with results in Figure 7 and supplementary figure v.

In this context, I am also wondering about the schematic in Fig. 4. The way I understand it, it shows simulated or even fictitious data/fits (although, in #700 it says "example"?). If the idea of this figure is to convince the reader of the new model fitting approach, wouldn't it be more appropriate to demonstrate the fitting procedure with real example neurons, e.g. the one from Fig. 5, and its shuffled control to demonstrate its sensitivity?

REPLY: Our goal was not to convince the reader: this method has been published many times before in refereed journals (Keith et al. 2009, DeSouza et al. 2011, Sajad et al. 2015, 2016, 2020; Sadeh et al. 2015, 2018, 2020; Bharmauria et al. 2020, 2021). Our goal was to explain the statistical method in an intuitive way. We therefore used simulations in Fig. 4b to exaggerate the effect, but everything else is real data.

ACTION: As requested, we have contrasted a fit to real data versus our shuffled data in Supplementary Figure iii (Page 13, lines 294-297; Page 60, lines 1341-1346). The shuffling mostly obliterates the 'hot spot' and increases the residuals of fit, as one would expect (Page 60, lines 1341-1346).

Interpretation of the results: In response to previous reviewer comments, the authors expanded on the argument that the results do not (necessarily) indicate a shift of the RF towards the landmark, i.e., not a spatial shift (#459). I still find that then statements like the one in #468 ("shifts toward TL") are potentially confusing - I guess what is meant with the "shift" here is not a spatial shift but that the TL model compared to the TF model gains more weight? If this meant then I find it still confusing that in the next sentence it is argued that the observed shift might explain gaze shifts - which would be indeed spatial shifts?

REPLY: The point here is that a response field fixed relative to the landmark would appear to shift relative to gaze, if the landmark shifted relative to gaze by the same amount and direction on each trial. However, in the current study this dissociation was produced by relatively small, pseudorandom variations in eye torsion from one trial to the next. This would not have a systematic influence on the overall response field. The point about influencing gaze came from our other studies (Bharmauria et al. 2020, 2021) where we looked at the influence of a larger shift in the landmark itself. This influenced memory and motor responses that in turn influenced gaze direction.

ACTION: This section has been re-written (Pages 28-29, lines 659- 665).

In general, I find the writing still worth improving; often terms and concepts are not well introduced and references to previous paragraphs/concepts/thoughts remain unspecific (see examples below), making it overall a hard read. The paragraphs in the introduction

about own previous work (added in response to previous reviewer requests) partly read like a narration (we did this, then we did that...) rather than a succinct line of argumentation and putting into context of the research question. Reviewer 1 had already complained in the previous review about writing style; especially the new paragraphs would benefit from thorough internal revisions; some specific examples of what I mean can be found below.

ACTION: Much of the paper has been edited / rewritten. To offset the new materials we added, many of the existing sections have been abbreviated (including the Introduction).

Specific points:

#67 "stabilizing noise" - does not make sense to me; did you mean "stabilizing against noise fluctuations" or something similar?

ACTION: The reviewer is right. We have corrected this (Page 3, line 64).

#96 "this process" - reference unclear (which process is meant?)

REPLY/ACTION: We meant the ego/allo weighting process. We have clarified this (Page 4, line 87).

#97 "this task" - again, reference unclear; please be specific

REPLY/ACTION: We have specified the task (cue-conflict, Page 4, line 91).

#100 "this landmark influence first appeared as a multiplexed signal in the memory response, and then became fully integrated into the final motor code" - sound a bit like lab-slang - "multiplexed" in what sense? what is meant by "motor code" (the explanation in parentheses does not help)?

REPLY: In this instance we were referring to a signal that contained uncorrelated egocentric and allocentric information. By motor code we meant the code in the perisaccadic motor burst.

ACTION: We have removed 'multiplexed' here and reworded the language (Page 4, lines 96-101). We later use 'multiplexing' specifically to describe our new result that FEF neurons can simultaneously encode target and landmark location information.

#123 "a relatively unprocessed [...] manner" - meaning unclear

ACTION: We have eliminated this.

#124 "to do this" - do what (previous sentence says "we hypothesized")?

ACTION: We have rephrased this part (Page 5, lines 111-120).

#134 "for more precise behavior" - is there a performance measure for motor accuracy that the authors are referring to? Was this introduced already?

ACTION: We have rephrased this part (Page 5, lines 119-120)

#150 "final gaze" - undefined; which time period is analyzed here?

REPLY: Final gaze refers to the gaze end point where the monkey's eye landed after the saccade was performed to the memorized target (Page 6, lines 135-136).

#150 "27%" - of what, the target-landmark distance ("15%" correspondingly)? Please specify.

REPLY: There was a systematic bias in both animals of the final gaze location 'toward the landmark'. It was already mentioned in the main text as:

The sentence: We found that there was a systematic bias (27 % for Monkey L and 15 % for Monkey V) of the final gaze location (gaze end point where the monkey's eye landed after the saccade was performed to the memorized target) toward the landmark, (Page 6, lines 134-139).

#255 "we also tested other" - please specify which other continua you tested

REPLY: that phrase was removed. The current paper now tests two continua.

#306 "variability [...] persists due to non-spatial factors such as attention and motivation" - speculative, rather a point of discussion than results

REPLY/ACTION: It's not a result but here we are reviewing describing the method, and people often ask why residuals don't go to zero. We have added the word 'likely' due (Page 13, line 293).

#324 the main text does not make reference to Fig. 5d-g (and the corresponding figure caption is generic). What am I supposed to learn from the results shown in these panels?

REPLY: Figure 5 does not show a result. It was just supposed to show how we divided the data by the four landmark configurations and that there was a reasonable distribution of data in each case for fitting.

ACTION: We have moved this to the supplementary section.

#353 "(Fig. 6c" - parenthesis missing

ACTION: This is corrected in the new Figure (7).

#705/712 "tests the explicit spatial codes used by neurons" - no, we do not know what neurons are actually using; it is just a fitting procedure.

ACTION: Rewritten (Page 41, lines 933-934): 'In order to determine what spatial frame best describes a response field, one must first confirm that the neuron has a spatially tuned response field.'

#715 please introduce PRESS

ACTION: We had already introduced PRESS in the section 'Fitting neuronal response fields against spatial models). However, as asked by the reviewer we have further clarified the fitting procedure using PRESS (Page 37-38, lines 856-862).

#736 "stabilized median" - unclear, median of what? Also, if the computations were done with higher number of shuffle iterations, as the sentence suggests, then why not show the result with the higher numbers, which would be statistically safer?

ACTION: Here, the stabilized median means that the cumulative median does not change (from one shuffling to the next) by more than half of our discretization (0.1). This is determined by calculating up to one thousand shuffles for all the data. For the analysis one hundred shuffles are used to keep the computing time manageable since the number of shuffles is multiplied with the number of bootstraps.

Caption Fig. 3: "more robust plots" - I guess you mean responses, not plots

REPLY/ACTION: Yes, the reviewer is right. We have replaced the 'plot' with 'mean responses' (page 10, lines 225-228).

Caption Fig. 6: It is not explained in which sense this is a swarmchart, since it looks like a regular scatter plots. More important, if both the x- and the y-axis are meaningful and the distribution width and deviations from unity line are main results then I do think that jittering the data for better visibility is justified.

REPLY/ACTION: Since many data points were overlapping with each other, we used a plot that minutely jitters the same data point to make it more distinguishable from the other overlapped data point. These charts are called swarm plot charts. Note that the figure numbers have changed now.,

Fig. 3 and 5/data sampling: Since the grand average curves in Fig. 3b and the corresponding example data (e.g. Fig. 5a) contain the top 10% of responses, one wonders if any response would be left, when looking at only the remaining 90% - probably not, but one cannot tell since the second curve contains also the top 10%; if the 90% "residuals" are flat, does it still make sense to include this (likely very noisy) data in the analyses?

REPLY: Yes, it is critical to fit the entire response field, not just the peaks. One cannot determine the shape of a mountain range only by looking at the tops of the mountains. One must consider the entire contour. In particular, our method is most sensitive to the slopes (Keith et al. 2009). The only reason we show the top 10% is because that is what most people show, after they map out the peak of the response field.

ACTION: This is mentioned in Methods, Pages 35-36 lines 819-821.

Fig. 6/7: Please do not overlap data with text boxes.

ACTION: We have corrected this.

Fig. 6: Why TL(C1) label in c/d, isn't this pooled data?

REPLY/ACTION: The figure is changed now, however, we have been more careful with labeling. It was a typo in the earlier version.

Reviewer #4 (Remarks to the Author):

This version reads significantly better than the previous version.

REPLY: We thank the reviewer for their review.

Some additional comments:

- Supp. Fig. iv: panel a: I still do not understand why the authors like to obscure data points with the color legend. The spikes are hidden by the color legend. Also, please explain in the figure caption whether the shown raster is for the top 10% of trials or whether it is for all trials.

ACTION: We have removed the colored legend to outside of the figure now.

- panels b-e: I'm also not sure what the label of the colorbar means. What does Normalised neural activity mean? By normalisation, did you baseline-subtract or what exactly? i.e. in what sense is it "normalised"? Please clarify

ACTION: This was a mistake. The color represents the non-parametric fit to the data points. We have now corrected the legend.

- Fig. 5: similar comments to the above (except the top 10% raster, which is now clearly described for this figure)

ACTION: We have now moved the legend outside and done the caption accordingly.

- Fig. 6a, b: again, the legends look like they may hide data points. Also, what does “signi units” and “not signi units” mean???

REPLY: Figure 6 is moved to Figure 7 now. The legend within the figure might have obscured the data points. We haven't kept any unnecessary label within the figure now. The label related to 'signi and non signi units' (meaning significant and insignificant unit respectively) has now been removed.

- there are still too many typos, missing parentheses etc for my taste

ACTION: We have extensively proofread the paper and performed spell check.

REVIEWERS' COMMENTS:

Reviewer #4 (Remarks to the Author):

Just a couple of minor comments:

- lines 55-56: I'm having a very hard time understanding the blue text. "Biggest shift" with respect to what?

- By the way, the authors should inspect their PDF's before "validating" for submitting. Both PDF's have huge amounts of missing text. Examples:

- lines 129-130: I see weird parentheses with no text in them or a parentheses with a space then ": a large `cross`". Both PDF's showed these weird things.

- similarly in the figure 1 legend (like 155)

I had to go back to the original source Word files to look if the text could be read more seamlessly. So, please inspect the submission PDF's in the future before validating. Thanks.

Reviewer #4 (Remarks to the Author):

Just a couple of minor comments:

- lines 55-56: I'm having a very hard time understanding the blue text. "Biggest shift" with respect to what?

REPLY: We have further clarified it as: "the biggest shift from eye-centered toward landmark-centered target coding" (page 2, line 58)

- By the way, the authors should inspect their PDF's before "validating" for submitting. Both PDF's have huge amounts of missing text. Examples:

- lines 129-130: I see weird parentheses with no text in them or a parentheses with a space then ": a large 'cross'". Both PDF's showed these weird things.

- similarly in the figure 1 legend (like 155)

REPLY: We carefully checked our submitted document, and we didn't find any such typos/errors in our document. These errors might have arisen due to some bug while the referee downloaded their document from the online portal. However, we thank the reviewer for patience. We have reviewed the document one more time.